# A structural and dynamic model for the assembly of Replication Protein A on single-stranded DNA

Luke A. Yates [1], Ricardo J. Aramayo[1], Nilisha Pokhrel[2], Colleen C. Caldwell[3], Joshua A. Kaplan[1], Rajika L. Perera[1,4], Maria Spies [3], Edwin Antony [2] & Xiaodong Zhang [1]

Replication Protein A (RPA), the major eukaryotic single stranded DNA-binding protein, binds to exposed ssDNA to protect it from nucleases, participates in a myriad of nucleic acid transactions and coordinates the recruitment of other important players. RPA is a hetero-trimer and coats long stretches of single-stranded DNA (ssDNA). The precise molecular architecture of the RPA subunits and its DNA binding domains (DBDs) during assembly is poorly understood. Using cryo electron microscopy we obtained a 3D reconstruction of the RPA trimerisation core bound with ssDNA (~55 kDa) at ~4.7 Å resolution and a dimeric RPA assembly on ssDNA. FRET-based solution studies reveal dynamic rearrangements of DBDs during coordinated RPA binding and this activity is regulated by phosphorylation at S178 in RPA70. We present a structural model on how dynamic DBDs promote the cooperative assembly of multiple RPAs on long ssDNA.

[1] Section of Structural Biology, Department of Medicine, Imperial College London, Sir Alexander Fleming Building, South Kensington, London SW7 2AZ, UK. [2] Department of Biological Sciences, Marquette University, Milwaukee, WI 53201, USA. [3] Department of Biochemistry, Carver College of Medicine, University of Iowa, Iowa City, IA 52241, USA. [4] Present address: Poseidon LLC, 2265 East Foothill Boulevard, Pasadena, CA 91107, USA. Correspondence and requests for materials should be addressed to X.Z. (email: xiaodong.zhang@imperial.ac.uk)

Replication Protein A (RPA) is an abundant multi-domain heterotrimeric protein complex essential to nearly all DNA processing events inside eukaryotic cells[1]. As the primary single-stranded DNA (ssDNA)-binding protein in eukaryotes, RPA acts to coat and protect exposed ssDNA from nucleases[2], as well as forming a physical platform to recruit other factors to the DNA including those involved in DNA damage signaling, DNA repair, and DNA replication. RPA is thus a critical component for replication, repair, and recombination pathways[3].

Comprised of three protein subunits, denoted as Rfa1, Rfa2, and Rfa3[4] (in *Saccharomyces cerevisiae*) or RPA70, RPA32, and RPA14 (in humans), RPA contains multiple oligonucleotide/oligosaccharide binding (OB)-folds that interact with both ssDNA and proteins (Fig. 1a)[5]. The largest subunit, Rfa1, possesses four OB domains: DNA-binding domain-A (DBD-A), B (DBD-B), C (DBD-C), and F (DBD-F), which are connected by flexible linkers. DBD-F, situated at the N-terminus also serves as a site of protein–protein interactions with other factors. Rfa2 possesses a single OB fold (DBD-D) followed by a Winged-Helix (WH) domain, which also participates in protein–protein interactions[6]. The smallest subunit Rfa3 has a single OB-fold (DBD-E) and forms part of the trimerisation core (Tri-C) consisting of DBD-C of Rfa1, DBD-D of Rfa2, and DBD-E of Rfa3. Although both DBD-F and DBD-E have also been shown to weakly interact with DNA, DBD-A, DBD-B, DBD-C, and DBD-D are primarily responsible for RPA's ssDNA-binding activities[7–9]. The

**Fig. 1** Cryo-EM image analysis of RPA-coated ssDNA. **a** Schematic of the RPA subunits Rfa1, Rfa2, and Rfa3 that form the RPA complex. Domains are shown and labeled with those associated in the trimerisation core indicated by dotted lines. **b** A typical cryo-EM micrograph of ScRPA-dT$_{100}$ nucleoprotein complexes. Different types of single particles analyzed are circled. The 2D class averages that result from the types of particles are shown in **b–d** and are colored by Tri-C (blue), dimer (pink), and trimer (green). Scale bar represents 50 nm. **c** 2D-class averages of individual RPA trimerisation core heterotrimers. Scale bar in **c–e** represents ~80 Å. **d** 2D-class averages focusing on two RPA molecules. Two trimerisation cores (Tri-C) of RPA are clearly distinguishable in these images. However, there is some conformational flexibility of the Tri-C relative to one another. In some classes it is also clear that there is a domain sandwiched between two RPA Tri-Cores. **e** 2D-class averages of ScRPA-dT$_{100}$ showing up to three molecules of RPA per oligonucleotide, which agrees with EMSA and SEC-MALS data (Supplementary Fig 2c, d)

Rfa3 subunit is essential for proper RPA function, with its deletion being lethal[10], but is thought to only provide a structural role within the complex.

Due to its modular nature, RPA is extremely flexible and its six DBDs can adopt multiple conformations. Structural and biochemical studies have focused on RPA (and domains thereof) associating with short fragments of ssDNA (up to 32 nt) using NMR[11,12], X-ray crystallography[9,13–17] and SAXS[18]. Structural, biochemical, and biophysical studies have shown that RPA can associate with ssDNA in different modalities; a low affinity mode which binds 8–12 nucleotides (nts), with structural studies showing that DBD-A and DBD-B can bind and cover 8 nts ssDNA[13,18–20], and a high-affinity compact mode involving all four major DBDs (A–D) that bends a 28 nts ssDNA tract in a horse-shoe shape configuration[13,21]. Based on these studies it was thought that the Tri-C, which contains DBD-C and DBD-D, has a weaker association with ssDNA compared to DBD-A and DBD-B and is only involved in binding to longer ssDNA in the compact mode[9,13,18,22].

However, within the context of the cell, RPA binds and protects much longer ssDNA tracts (many kilobases), such as end resection intermediates during homologous recombination (HR). Furthermore, recent studies suggest that not only do RPA molecules form dynamic complexes on ssDNA but also undergo diffusion[23,24], revealing a much more complex picture of RPA on ssDNA. Interestingly, early electron microscopy work of RPA on long ssDNA substrates (several kilobases) revealed that yeast RPAs form nucleoprotein complexes that are organized into nucleosome-like particles and do so with high cooperativity[25]. However, human and fruit fly RPAs have been shown to bind ssDNA with low cooperativity[5,26,27]. Additional studies by EM and atomic force microscopy (AFM) also reveal a diverse range of conformations of RPA-coated ssDNA, such as amorphous condensates, beads-on-a-string, or extended arrangements[28–30]. Thus, RPA does not appear to form highly ordered nucleoprotein filaments, a property displayed by other ssDNA-binding proteins, such as SSB, RecA, and RAD51, and indeed there is very little information on how multiple RPA molecules bind to ssDNA and whether there is any coordination between RPA molecules.

To better understand the assembly of RPA molecules on long DNA substrates, we undertook structural and ensemble analysis of multiple RPA molecules. In this study, we obtained structures of single and multiple RPA molecules on ssDNA using cryo electron microscopy (cryoEM) and our results provide new insights into how RPA interacts with ssDNA. Combined with Förster resonance energy transfer (FRET) experiments, we uncover an unanticipated mechanism of RPA self-association that provides a potential mechanism for its loading, exchange, and remodeling on long ssDNA tracts.

## Results

**Structure of *S. cerevisiae* RPA-ssDNA.** We produced recombinant full-length *S. cerevisiae* RPA heterotrimer complexes of Rfa1, Rfa2, and Rfa3 in bacteria (denoted as ScRPA, Supplementary Fig. 1). Recombinant ScRPA was purified to homogeneity as judged by SDS–PAGE and gel filtration analysis coupled with in-line multi-angle laser light scattering (SEC-MALS), suggesting intact complexes (Supplementary Fig. 1a, b). We used a 100 nt polypyrimidine oligonucleotide to form a RPA-ssDNA complex containing multiple RPA molecules (dT$_{100}$ and Supplementary Fig. 2a, b). Occluded site sizes for ScRPA vary from 18 to 25 nt depending on solution conditions[31] and thus up to four RPA molecules are expected to bind to a dT$_{100}$ substrate. Electrophoretic mobility shift assays (EMSA) together with SEC-MALS of ScRPA-dT$_{100}$ complexes confirm 3–4 RPA molecules

associated to ssDNA (Supplementary Fig. 2c, d). We subsequently collected cryoEM data of the purified ScRPA-dT$_{100}$ complexes and subjected the images to single particle analysis (Fig. 1 and Table 1). Within the datasets we obtained several reconstructions, with multiple ScRPA heterotrimers coating ssDNA, as well as individual ScRPA-ssDNA complexes (Fig. 1b–d).

Combining two datasets and several rounds of 2D and 3D classification (Supplementary Fig. 3a) we isolated a fraction of the particles that enabled a 3D reconstruction corresponding to the ScRPA Tri-C associated with ssDNA to a global resolution of 4.7 Å (Fig. 2a, b and Supplementary Fig. 3b–d), a surprisingly high-resolution structure by cryoEM given the small size of the protein–nucleic acid complex (~55 kDa with ssDNA). Although full-length proteins were used to form the ScRPA-ssDNA complexes, (Supplementary Fig. 1), the winged-helix domain, as well as some of the N-terminus of Rfa2 were invisible. However, this was anticipated based on previous observations[17]. More strikingly, the N-terminal half of Rfa1 (DBD-F, DBD-A, DBD-B) is not visible in this reconstruction.

The reconstruction measures ~81 Å × ~58 Å × ~36 Å in dimensions and a model of the ScRPA Tri-C (based on homology models) can be fitted into the EM density map. Subsequently, the model was flexibly fitted using molecular dynamics flexible fitting (MDFF)[32,33] and further refined using jelly-body refinement routines in Refmac5[34] within CCPEM[35], and real-space refinement procedures in Phenix[36]. At this resolution, the boundaries

**Table 1 Cryo-EM data collection, refinement, and validation statistics**

|  | ScRPA-dT$_{100}$ (EMDB-4410) (PDB 6I52) | |
| --- | --- | --- |
| **Data collection and processing** | **Dataset 1 (WT)** | **Dataset 2 (S178D)** |
| Magnification | 130,000 | 130,000 |
| Voltage (kV) | 300 | 300 |
| Electron exposure (e-/Å$^2$) | 49 | 50 |
| Defocus range (μm) | −1 to −3.5 | −1 to −4 |
| Pixel size (Å) | 1.055 | 1.06 |
| Symmetry imposed | C1 | C1 |
| Initial particle images (no.) | ~1,300,000 (Tri-C), ~200,000 (dimer) | |
| Final particle images (no.) | 341,873 (Tri-C), 32,583 (dimer) | |
| Map resolution (Å) | 4.7 (Tri-C), 7.5 (dimer) | |
| FSC threshold | 0.143 | |
| Map resolution range (Å) | 4.5–6 (Tri-C), 6–10 (Dimer) | |
| **Refinement** | **Tri-C** | |
| Initial model used (PDB code) | Homology model derived from 1l1o and 4gop | |
| Model resolution (Å) | 4.7 | |
| FSC threshold | 0.143 | |
| Model resolution range (Å) | 4.5–6 | |
| Map sharpening B factor (Å$^2$) | −400 | |
| Model composition | | |
| Non-hydrogen atoms | 3856 | |
| Protein residues | 3456 | |
| Ligands | 400 | |
| R.m.s. deviations | | |
| Bond lengths (Å) | 0.28 | |
| Bond angles (°) | 0.60 | |
| Validation | | |
| MolProbity score | 2.71 | |
| Clashscore | 9.67 | |
| Ramachandran plot | | |
| Favored (%) | 87.7 | |
| Allowed (%) | 11.4 | |
| Disallowed (%) | 0.9 | |

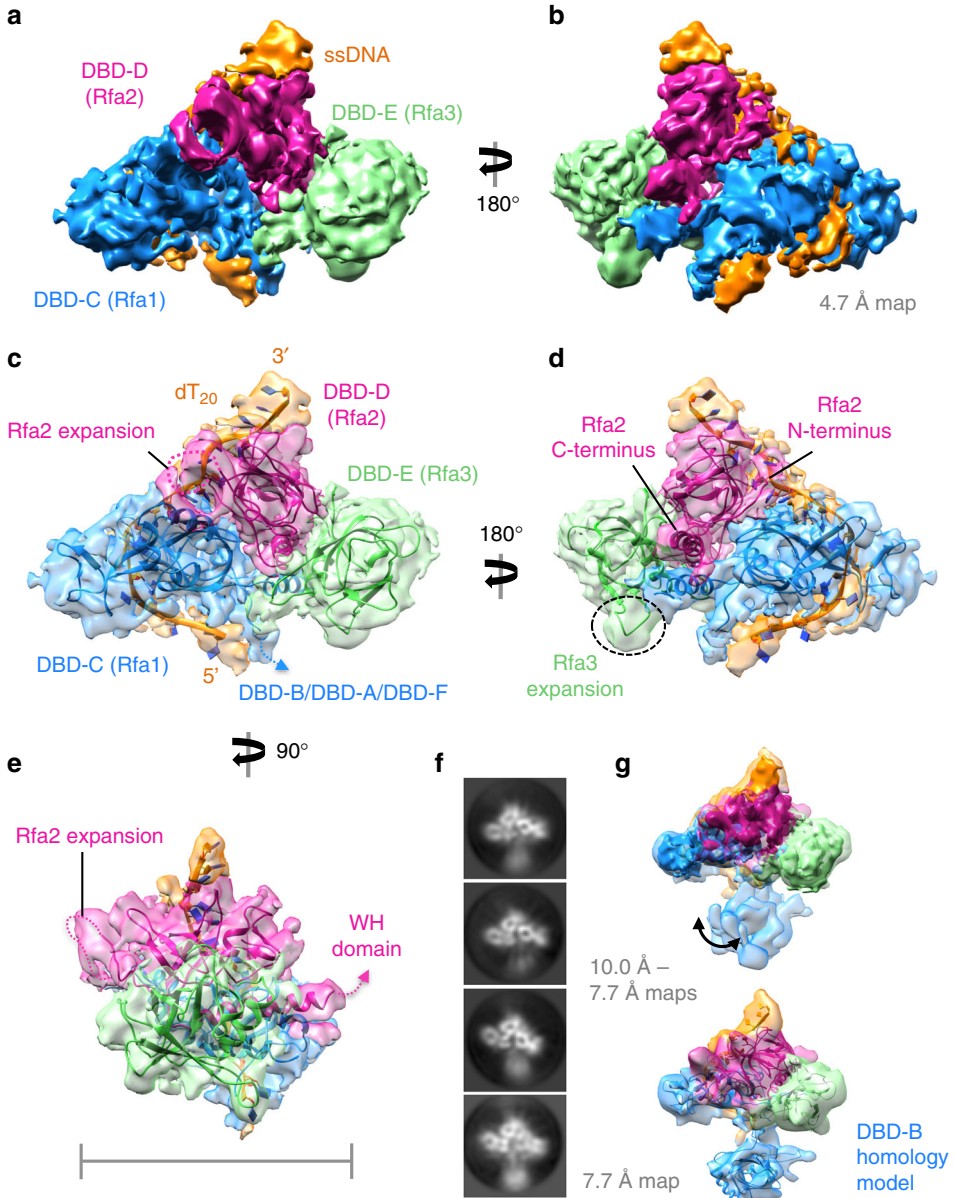

**Fig. 2** Cryo-EM structure of ScRPA trimerisation core. **a, b** Cryo-EM reconstruction of ScRPA trimerisation core determined to 4.7 Å resolution by gold standard-FSC, post-processed and sharpened in Relion and colored by subunit; Rfa1 in blue, Rfa2 in pink, Rfa3 in pale green, and poly-dT in orange. **c–e** Orthogonal views of a fitted and refined homology model of ScRPA Tri-C. The subunits are colored as in **a** and a number of yeast-specific features in Rfa2 and Rfa3 that are expansions of the amino-acid polypeptide chain protrude and from loops that are visible in the cryo-EM map. The polypeptide chain linkers (missing in the structure) to the C-terminal winged-helix domain and N-terminal phosphorylation domain of Rfa2 are labeled, as well as the DbdB-DbdA and N-terminal domain of Rfa1. Scale bar represents ~70 Å. **f** Representative 2D-class averages showing an additional domain underneath the Tri-C that is positionally flexible. **g** Three individually refined maps from sub-classification of the data at sub-nanometer resolution, by gold standard-FSC (see Supplementary Fig 4), aligned on the Tri-C and fitted with 4.7 Å Tri-C map showing an additional small domain and its different locations in each map. The apparent domain motion is arrowed. Below, the highest resolution map (7.7 Å) with fitted atomic model from **c** showing that the additional density region is sufficient to fit an OB-fold. A homology model for DbdB is fitted and shown. Figures were generated in UCSF Chimera[56]

of the three subunits could be identified (Fig. 2c–e). There are a number of yeast-specific insertions compared to the human or fungal structures, the most obvious being for DBD-D and DBD-E (Fig. 2c–e). The 14 amino acid flexible expansion in DBD-D, for which we see only partial density, has no known function in DNA replication or repair when removed, despite it possessing a phosphorylation site for Mec1 (yeast orthologue of human ATR), the major PI3K kinase in DNA damage signaling and replication[37,38]. The ssDNA path associated with Tri-C, as observed in the crystal structure of *Ustilago maydis*

RPA-ssDNA[13], fits into the additional density surrounding the RPA Tri-C (Fig. 2c, d, orange) and covers ~20 nucleotides. The presence of ssDNA in the reconstruction is further supported by the observation that in the absence of ssDNA, RPA molecules do not assemble into dimers and trimers and the majority of the Tri-C particles were isolated from dimers and trimers.

In addition, we have obtained several sub-nanometer resolution reconstructions of the Tri-C with an additional domain in varying locations/orientations (Fig. 2f, g and Supplementary Fig. 4a–c). This additional density is sufficient for a single

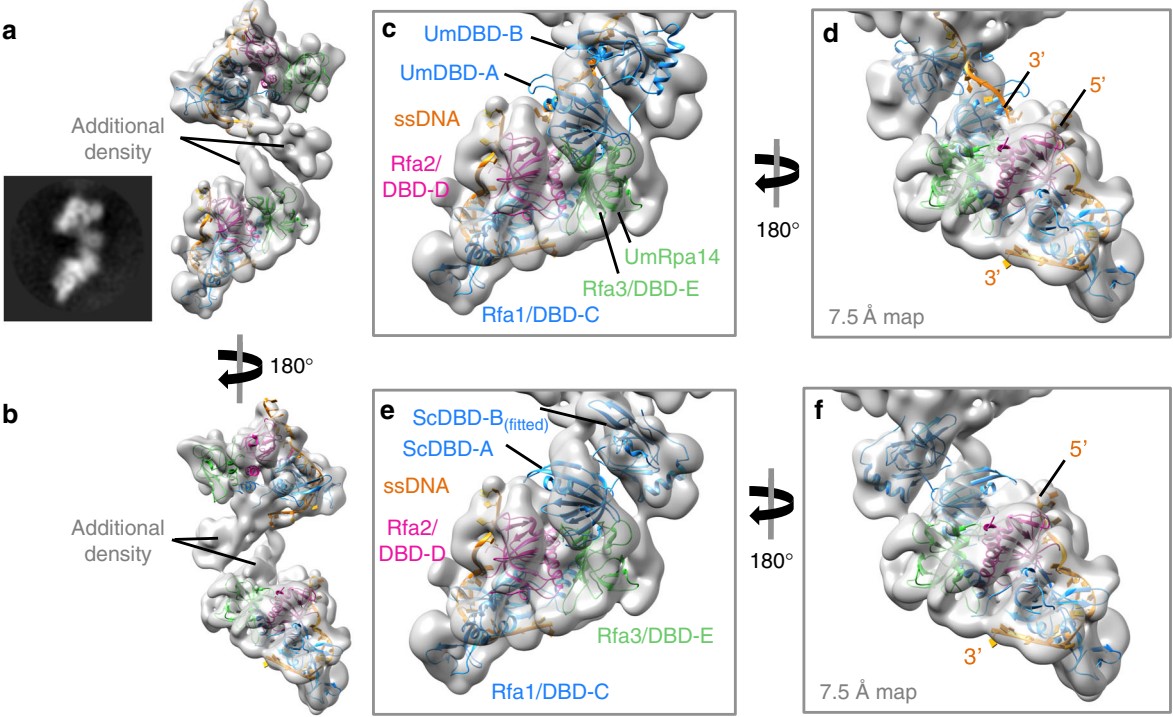

**Fig. 3** CryoEM and structural analysis of an RPA dimer on ssDNA. **a**, **b** 3D reconstruction of yeast RPA dimer subclass at sub-nanometer resolution (7.5 Å by gold standard-FSC). The models fitted in the 4.7 Å maps (colored as in Fig. 2) are fitted into the density corresponding to the Tri-C. Additional density not described by the fitted of the models. A 2D class average of a similar view is shown alongside (**a**). **c**, **d** The atomic co-ordinates Rpa14, DbdA/DbdB, and the associated ssDNA were excised from pdb: 4GOP and aligned by secondary structure onto DBD-E. The positions of DBD-A and DBD-B, relative to Rpa14 (aligned on DBD-E), are in similar positions to the additional density unaccounted for by either Tri-C. **d** The 5′ and 3′ ends of the ssDNA in each model are labeled to indicate that ssDNA could pass from DBD-A to the Tri-C. **e**, **f** DBD-A and DBD-B of the fungal structure from **c**, **d** are replaced with the yeast DBD-A NMR structure (pdb 1YNX) and a Phyre2-generated DBD-B homology model by superposition. DBD-B, which is slightly out of the density, is adjusted slightly by fitted as a rigid body in Chimera. The density for DBD-A is slightly weaker than DBD-B, suggesting that even in this subclass there is some occupancy heterogeneity

OB-fold (Fig. 2g and Supplementary Fig. 5a–d), which we propose to be DBD-B because it shows connectivity to the Tri-C (Fig. 2). Comparisons with the compact 30-nt RPA crystal structure[13] (Supplementary Fig. 5e–g) reveal that DBD-B (and thus DBD-A) in our reconstruction are positioned differently from those in the crystal structure, suggesting that ssDNA follows a different path in our reconstruction compared to those in the compact mode observed in the crystal structure. FRET studies using labeled ssDNA supports our EM analysis and show that the ssDNA within a single ScRPA adopts an extended configuration (Supplementary Fig. 5h and i). Our studies suggest that DBD-A/DBD-B may be more dynamic than previously thought, whereas the trimerization core is more stable on ssDNA. This is consistent with single-molecule studies of DBD dynamics on ssDNA[39].

**ScRPA can self-associate on ssDNA**. 2D classifications suggest that two ScRPA Tri-C molecules seem to contact one another via an additional density region, forming an ScRPA dimer (Fig. 1c, d). However, there is an inherent degree of conformational flexibility between one Tri-C relative to another Tri-C and this would compromise the accuracy of image alignment and, in turn, limit the resolution of our 3D reconstruction of the ScRPA dimer on ssDNA. 3D classification reveals the ScRPA dimer architecture with varying degrees of connectivity/flexibility (Supplementary Fig. 6). Using a subset of dimeric ScRPA particles we obtained a dimeric RPA reconstruction at 7.5 Å overall resolution (Supplementary Fig. 6) and were able to fit the higher-resolution ScRPA Tri-C density and the model into the corresponding density in the

dimer reconstruction. There are additional density regions between the two fitted maps (Supplementary Fig. 6, Fig. 3a, b), suggesting an interaction involving DBD-E, which connects the two RPA molecules together (Fig. 3a, green subunit). However, the resolution of the map does not permit accurate modeling. We noted that in the crystal structure of the fungal RPA-ssDNA complex (pdb 4GOP), DBD-E of one molecule contacts DBD-A of another molecule through crystal packing[13] (Supplementary Fig. 7a, b). Using the atomic coordinates that encompass the DBD-E(RPA14) of one RPA molecule and the associated DBD-A and DBD-B from the neighboring molecule in the crystal packing, we could place this model into our ScRPA 'dimer' reconstructions (Supplementary Fig 7c) by aligning on the DBD-E models. DBD-A and DBD-B now fall into the unassigned density regions of the dimer (Fig. 3c, d). This also suggests how the ssDNA can pass from the Tri-C of one RPA to DBD-A of the adjacent RPA whilst maintaining polarity (Fig. 3c, d). We then replaced the fungal DBD-A and DBD-B domains with the yeast DBD-A (pdb 1ynx) and a DBD-B homology model. DBD-B has to be readjusted slightly to optimize its fitting into the density (compare Fig. 3c, e, with d, f), suggesting that DBD-B is flexible relative to DBD-A, in agreement with previous studies[19]. Comparison of the Tri-C-DBD-B map (Fig. 2g) with the dimer reconstruction (Fig. 3a) suggests that the density proposed to be DBD-B in Fig. 2g occupies similar locations as the assigned DBD-B in the dimer reconstruction (Fig. 3a, c). It is worth noting that some 2D class averages aligned on the lower Tri-C show a poorly defined Tri-C of the adjacent molecule (Supplementary Fig 6e). This further supports the relative flexibility between DBD-A and DBD-B

within the same RPA and multiple conformations exist between DBD-A and DBD-B. Such remodeling of DBD-A when multiple RPA molecules are bound on ssDNA is observed in FRET studies (shown below).

Our results, compared with previous crystal structures of RPA-ssDNA, thus suggest that within one RPA there is significant intrinsic flexibility between DBD-B and Tri-C, via the BC linker and again between DBD-A and DBD-B via the AB linker. Importantly, our data show a new interaction between DBD-E and DBD-A, which connect two ScRPA molecules and suggest a path for ssDNA to pass from one RPA to another.

**ssDNA becomes linearized when loaded with ScRPA**. Our structures and analysis suggest that the path of ssDNA within one ScRPA is different to a compact horse-shoe model. In the context of the dimer (or trimer) the ssDNA could be passed from one ScRPA Tri-C to another ScRPA Tri-C via DBD-A/DBD-B of one RPA interacting with DBD-E of another molecule, thus maintaining polarity and nucleolytic protection (Supplementary Fig. 7d, e). In order to investigate the structures and conformations in solution, we used Cy5 and Cy3 labeled ssDNA and FRET-based analysis to evaluate the configuration of ssDNA when single and multiple RPAs bind. We used $dT_{30}$ oligonucleotide labeled at the two termini with the Cy3 (FRET donor) and Cy5 (FRET acceptor) to monitor binding of single RPA (Fig. 4a). If the configuration of ssDNA is bent, like that observed in the crystal structure[13], we would observe a high FRET state. Upon binding of ScRPA, the FRET signal decreases in an RPA concentration-dependent manner and plateaus at a 1:1 ratio of RPA: $dT_{30}$ molecule (Fig. 4a). This suggests that the fluorophores are moved further away from one another and that ssDNA adopts a linear configuration, consistent with the ssDNA configuration suggested by our dimeric RPA-ssDNA cryoEM model. Similarly, binding of ScRPA to $dT_{90}$ oligonucleotides with internally positioned dyes (Fig. 4b, c and Supplementary Fig 8) results in the FRET decrease similar to that of the $dT_{30}$, which saturates at the 3:1 ratio of RPA: $dT_{90}$ and suggests equal extension of the RPA-saturated ssDNA molecule within the first RPA and between the two adjacent RPA molecules, fully consistent with our biochemical and structural studies.

**An association between DBD-A and DBD-E**. In order to confirm the specific DBD-A–DBD-E interactions at the dimer interface, we quantified these interactions between purified domains using microscale thermophoresis (MST). We produced recombinant DBD-A (Supplementary Fig 9) and DBD-E, by removing the C-terminal trimerisation helix (Supplementary Fig. 10), and purified these domains to homogeneity. Whilst DBD-A has already been produced in isolation[12], we used circular dichroism to ensure that the purified DBD-E retained its secondary structure (Supplementary Fig. 10g, h). Our MST data show that DBD-A can interact with DBD-E with an affinity of ~100 μM (Fig. 5a, b) as compared to ~0.3 μM for ssDNA ($dT_7$) in the same assay conditions. To validate the interaction interface, we made a number of charge-reversal point mutations at the proposed interface that are either highly conserved or invariant in other yeast species (Supplementary Fig. 11), and subsequently examined their effect on the DBD-E–DBD-A interaction by MST (Fig. 5 and Supplementary Fig. 10e, f). We found that the R78E, L88D, and R60E mutant DBD-E domains possess significantly weaker binding to DBD-A as compared to 'wild-type' (Fig. 5c–e). Whilst the MST experiments were performed in the presence of BSA (see Methods) to circumvent non-specific binding, a binding experiment with a BSA control also confirms that this interaction, although weak, is specific (Fig. 5f).

**An RPA phosphomimetic mutant promotes cooperative ssDNA binding**. Phosphorylation of Rfa1 at S178 has been observed in response to DNA damage and during mitosis and is highly Mec1-dependent[38,40–43]. An equivalent site in human RPA70 (T180) is also phosphorylated in an ATM and ATR-dependent manner[44]. Despite the conservation of this phosphorylation event, not much is known about how this modification alters RPA function. S178 is in close proximity to DBD-A and sits at the edge of the putative RPA–RPA interface (Fig. 5a). Therefore, we hypothesized that S178-phosphorylation might alter the dynamics of higher-order RPA assemblies on ssDNA via an altered RPA–RPA association. We produced a phosphomimetic S178D mutant DBD-A domain and tested its interaction with DBD-E via MST (Fig. 6a). We found that DBD-$A^{S178D}$ interacts with the DBD-E with an ~three-fold increased affinity ($EC_{50}$~30 μM) as compared to wild-type (~100 μM). We also found that the introduction of S178D substitution did not affect the isolated DBD-A's interaction with ssDNA ($dT_7$) (Supplementary Fig. 12a, b). Additionally, the presence of short oligos ($dT_{10}$) did not influence the DBD-A–DBD-E interaction of either the wild-type or S178D mutant DBD-A (Supplementary Fig. 12c, d), confirming that the substitution specifically affects interactions between DBD-E and DBD-A.

We reasoned that since DBD-E and DBD-A interactions mediate RPA–RPA dimer interactions, this phosphomimic might affect multiple RPA assembly on ssDNA. We therefore introduced an S178D substitution into ScRPA (denoted as RPA$^{S178D}$), and purified this mutant RPA complex analogously to wild-type to examine the effect of phosphorylation on long ssDNA ($dT_{100}$) binding (Fig. 6b, c). We analyzed the RPA$^{S178D}$-$dT_{100}$ by cryo-EM anticipating an improved dimer reconstruction due to increased affinity. Surprisingly, 2D class averages show that although one Tri-C is aligned very well and shows distinct features as expected (Supplementary Fig. 13), the density corresponding to a second Tri-C from an adjacent RPA is poorly defined, even compared to the 2D classes of the wildtype protein (Supplementary Fig. 13). This suggests that this RPA–nucleoprotein complex possesses greater flexibility. Investigating this mutant by FRET on $dT_{90}$ showed that, whilst the configuration of the ssDNA bound by RPA$^{S178D}$ is similar to wild type, it requires higher concentration of mutant proteins to saturate the ssDNA (Supplementary Fig 14). Consistent with this observation we found by MST that the phosphomimetic mutant possessed an apparent lower affinity ($EC_{50}$ ~80 nM) for $dT_{100}$ than wild-type (RPA$^{WT}$, $EC_{50}$ ~20 nM) on the same substrate. However, plotting the binding data as a linear Hill plot shows that the Hill coefficient is increased from ~1 for wild-type to 1.7 for ScRPA$^{S178D}$ (Fig. 6d), which is also supported by FRET data (Supplementary Fig. 13), suggesting cooperative binding by the S178D mutant. Given that this mutation does not affect the association of the individual domain to ssDNA (Supplementary Fig. 12a, b), the reduced affinity and increased cooperativity suggest that the DBD-A–DBD-B domains of one RPA might be partially displaced from ssDNA upon interacting with DBD-E of adjacent molecule.

To investigate whether DBD-A is indeed altered when multiple RPA molecules bind in a cooperative manner, we carried out FRET experiments using fluorescent versions of RPA. Since RPA binds with defined 5′–3′ polarity with DBD-A aligned closer to the 5′ end and DBD-D closer to the 3′ end, DBD-A from one RPA will be situated close to DBD-D of an adjacent RPA, when two RPA molecules are bound. Using non-canonical amino acids, we generated site-specifically labeled versions of RPA where DBD-A was labeled with Cy3 (RPA–DBD-$A^{Cy3}$) or DBD-D was labeled with Cy5 (RPA–DBD-$D^{Cy5}$). RPA–DBD-$A^{Cy3}$ were mixed with RPA–DBD-$D^{Cy5}$ and the change in

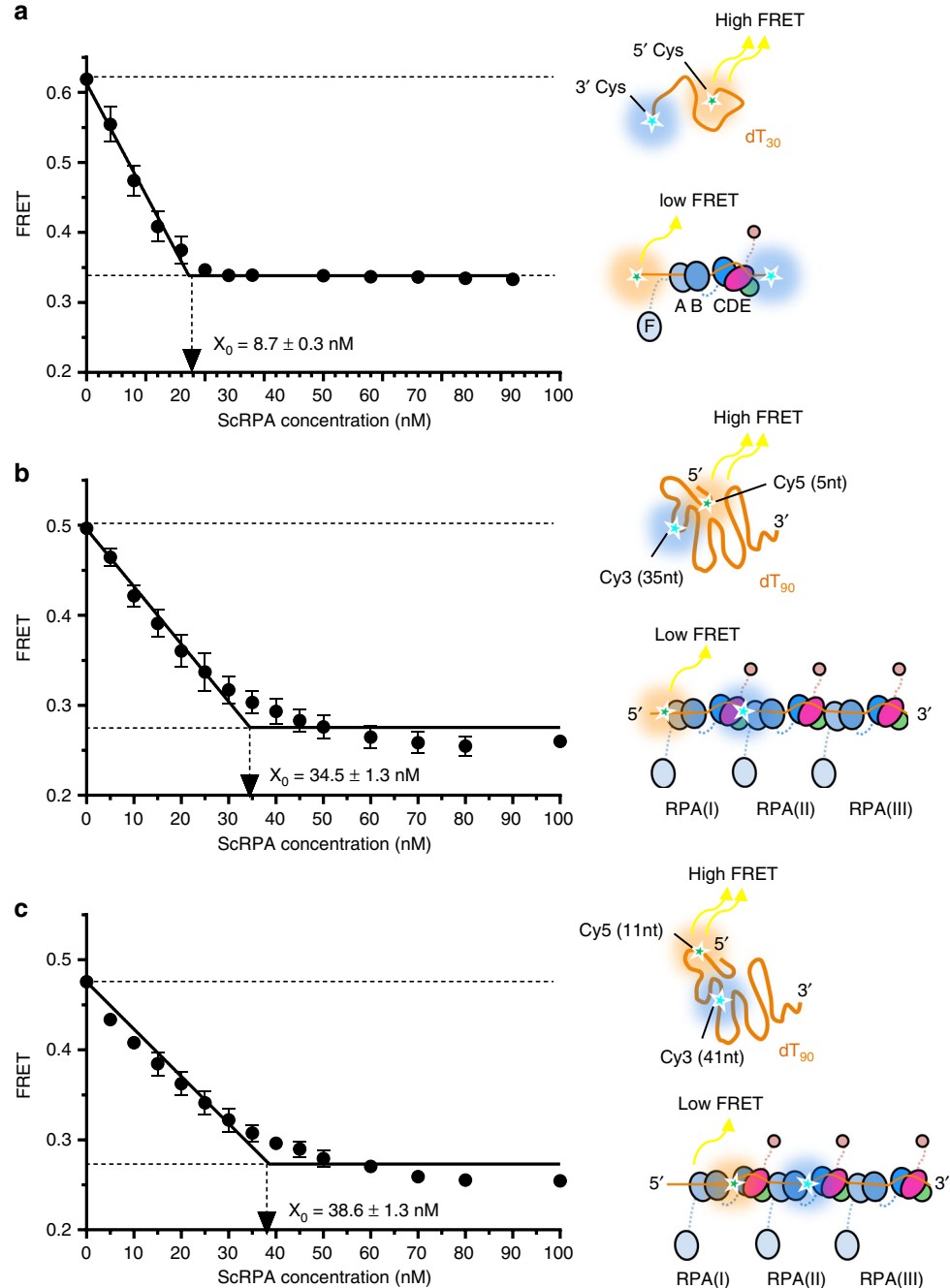

**Fig. 4** FRET-based analysis of the RPA-ssDNA complex formation. **a** Indicated concentrations of ScRPA were added to the solution containing 10 nM $dT_{30}$ oligo decorated with the Cy3 dye (FRET donor) at the 3' and the Cy5 dye (FRET acceptor) at the 5' end. Free $dT_{30}$ yields a high FRET signal due to the close proximity of the two dyes. Binding of ScRPA straightens the ssDNA thereby increasing the Cy3–Cy5 distance, which is manifested in the reduction in FRET. Under our experimental conditions, binding is stoichiometric with FRET decreasing linearly until the saturation is reached at approximately one ScRPA heterotrimer per one molecule of $dT_{30}$ ($X_0 = 8.7 \pm 0.3$ nM). The experiment was repeated twice and data points are an average with standard deviation (SD). **b**, **c** Indicated concentrations of ScRPA were added to the solution containing 10 nM $dT_{90}$ oligo decorated with the Cy3 and Cy5 dye separated by 30 nucleotides. In $dT_{90}$(5–35) Cy5 is 5 nt from the 5' end and Cy3 is 35 nt from the 5' end (**b**), whilst $dT_{90}$(11–41) Cy5 is 11 nt from the 5' end and Cy3 is 41 nt from the 5' end (**c**). In both cases the saturation of the FRET signal is achieved at about three ScRPA molecules per $dT_{90}$. Due to the dye positions, $dT_{90}$(5–35) reports on the ssDNA geometry within the RPA most proximal to the 5' end, while $dT_{90}$(11–41) reports on the ssDNA arrangement between the TriC of the first RPA and the DBD-A and DBD-B of the second RPA. In **b**, **c** the experiments were repeated three times and data points are an average with SD. In all data plotted, a cartoon is shown to illustrate the FRET experiment setup and the low and high FRET states observed. Source data are provided as a Source Data file

FRET was monitored as the RPA molecules formed cooperative complexes on increasing lengths of ssDNA in a stopped flow experiment (Fig. 6e). Structural models suggest that the high FRET signal can only arise from close proximity of DBD-A of one

RPA with DBD-D (Tri-C) of an adjacent molecule (Fig. 6f). Indeed, stopped-flow fluorescence data (Fig. 6g) shows the production of a FRET signal in the presence of ssDNA and we observe three key features.

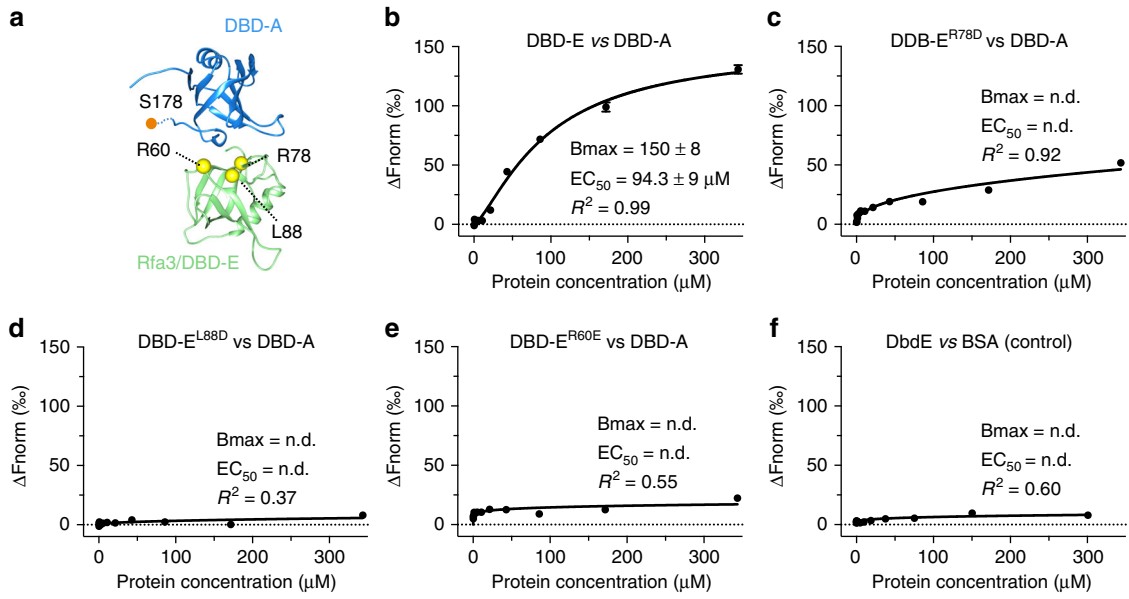

**Fig. 5** Investigation of the DBD-A–DBD-E interaction. **a** A proposed model for the interaction between Rfa1-DBD-A (blue) and Rfa3 (green) based upon the interaction interface between *Ustilago maydis* DBD-A and Rfa3 in the crystal structure asymmetric unit. The model was created using the yeast DBD-A NMR structure (pdb 1ynx) and the refined Rfa3 model from our cryo-EM data, superposed onto DBD-A and Rfa3 of pdb 4gop. Residues R78, L88, and R60 that we mutate in this study are located at the interface and are illustrated by yellow spheres. A Mec1-dependent DBD-A phosphorylation site (S178) close to the interaction interface is labeled. **b–e** Microscale thermophoresis (MST)-binding curves using purified DBD-A versus fluorescently labeled Rfa3-OB domain (residues 1–99) together with binding curves for the Rfa3-OB point mutants (substitutions given above curves). Normalized fluorescence was calculated using NanoTemper Analysis 1.2.101 and is plotted as function of DBD-A concentration. The apparent effective half maximal binding ($EC_{50}$) was calculated by fitting using the hill equation in Prism and is given for each binding experiment together with the calculated maximum fluorescence (Bmax) and $R^2$ of the fit. n.d. denotes not determined. **f** Bovine serum albumin (BSA) control for non-specific binding is also shown. Each experiment was performed in triplicate with each data point an average with standard error of the mean (SEM) shown. Source data are provided as a Source Data file

FRET analysis using a fixed concentration of ssDNA (40 nM) but varying oligonucleotide lengths ($dT_n$: $n = 20$–140 nt) with fluorescent versions of either wild-type RPA or $RPA^{S178D}$ show that the fluorescence changes only occur when RPA associates with ssDNA sufficiently long enough to accommodate two heterotrimers (≥45 nt) (Fig. 6g, h). This is consistent with the formation of RPA dimers bound to ssDNA. Given that the DBD-A–DBD-E affinity is relatively weak, it is therefore not unexpected that we do not observe FRET with RPA alone or with short oligos. From the stopped flow FRET analysis, we could also infer cooperative binding of $ScRPA^{S178D}$ by calculating the binding dynamics as a function of ssDNA length. The data could be fit with a two-step model[39], where the first step ($k_{obs,1}$) reflects a rate of association between RPA ssDNA and/or RPA heterotrimers. We find that the rate of association ($k_{obs,1}$) of the S178D mutant increases with oligonucleotide length (Fig. 6i and Supplementary Fig. 15a). However, the rates of association for wild-type RPA remain constant irrespective of ssDNA length (Fig. 6j and Supplementary Fig. 15b). This suggests that RPA–RPA complexes are promoted for the $RPA^{S178D}$, consistent with the domain interaction data, and indicate cooperative ssDNA binding.

Finally, the FRET intensity signatures shed light on the positioning of the DBDs. When wild type RPA is in excess (i.e., there is not enough binding sites for all DBDs) there is a rapid increase in signal followed by a decrease to a lower FRET state, suggesting remodeling (Supplementary Fig. 14c, d). Similar experiments with $RPA^{S178D}$ show a monophasic increase in FRET signal that is stable consistent with an enhanced DBD-A–DBD-E interaction (Supplementary Fig. 14c). Additional experiments on dT140 show a difference in FRET signatures between the WT and S178D that could arise from the differences in flexibility/mobility

or reduced ssDNA coverage of the $ScRPA^{S178D}$ we observe by cryo-EM (Supplementary Fig. 14c, d). These data are consistent with a model whereby the interaction between two RPA molecules is enhanced by phosphorylation at S178.

In summary, our combined data indicate that $ScRPA^{S178D}$ mutant has enhanced affinity with adjacent RPA on ssDNA through DBD-A and DBD-E interactions, which result in weaker association of DBD-A and DBD-B with ssDNA. The implications of this property are that RPA might be more easily displaced and ssDNA is more exposed. Indeed we find that $ScRPA^{S178D}$ nucleoprotein complexes can be more readily exchanged in vitro by another ssDNA-binding protein RecA compared to wild-type (Supplementary Fig. 16a–d) and the $ScRPA^{S178D}$-$dT_{100}$ is slightly more sensitive to S1 nuclease digestion (Supplementary Fig. 16e, f).

## Discussion

A crucial feature of RPA is that, whilst being able to bind nucleic acids with very high affinity, it must also be easily displaced to 'hand-off' ssDNA to other enzymes for further downstream processing. Of the six OB-folds of RPA, four possess the predominant ssDNA-binding activity and, when all engaged, allow RPA to associate with ~30 nt with a Kd of ~0.05 nM that could adopt a compact horse-shoe structure[9,13,21,22,31]. Individual OB-domains possess differing affinities for nucleic acids with DBD-A and DBD-B considered the high-affinity-binding domains (~50 nM) and the remainder of the RPA complex (DBD-C, DBD-D, and DBD-E), which comprise the Tri-C, thought to have much weaker ssDNA-binding activity (low micromolar). Although it should be noted that these affinity measurements were obtained using biochemically isolated DBDs, the DNA-binding properties

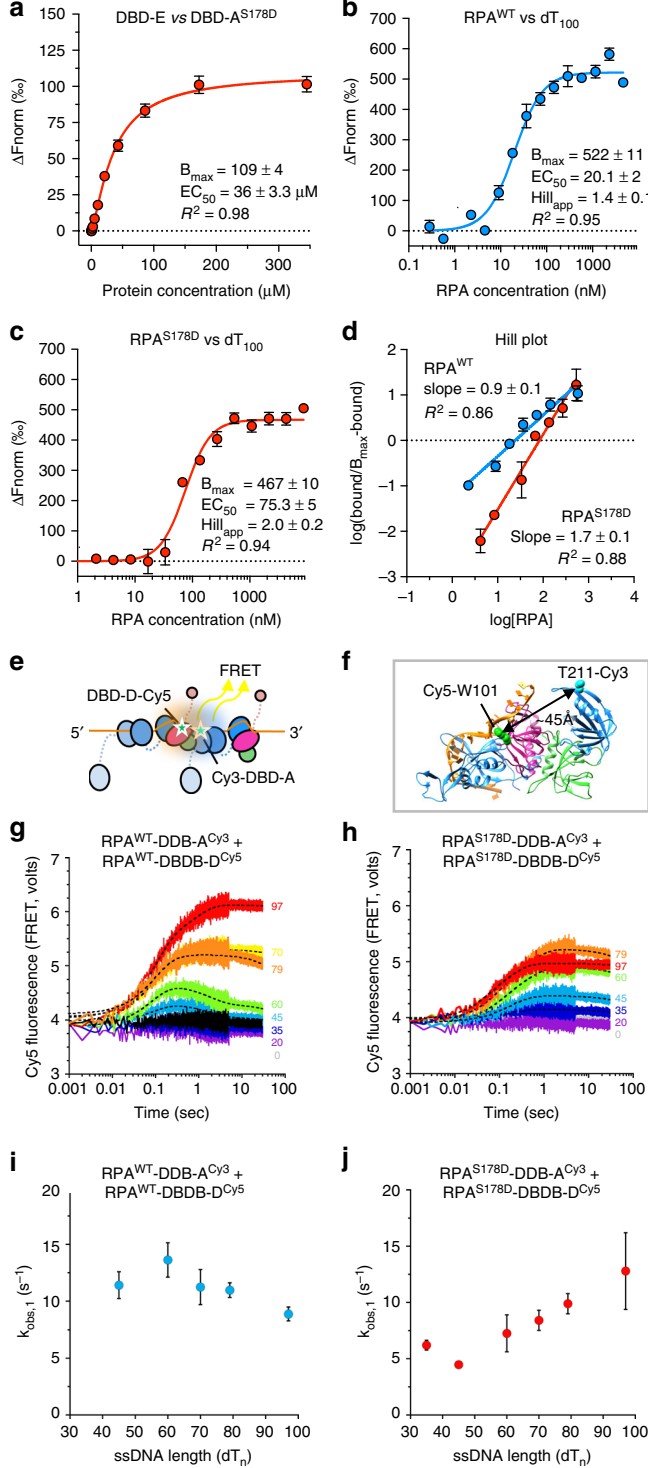

**Fig. 6** Biophysical assessment of an Rfa1 phosphomimetic mutant.
**a** Microscale thermophoresis (MST) binding curves of DBD-A$^{S178D}$ mutant versus Rfa3-OB. **b**, **c** MST-binding curves of RPA heterotrimeric complexes interacting with ssDNA, using wild-type RPA (red, RPA$^{wt}$) and RPA phosphomimetic mutant S178D (blue, RPA$^{S178D}$) and a Cy5-labeled dT$_{100}$ ssDNA substrate. For all MST curves, apparent EC$_{50}$ values were calculated using the Hill-equation. Normalized fluorescence was calculated using NanoTemper Analysis 1.2.101. Measurements were repeated five times and data points are averages shown with SEM, maximum fluorescence (Bmax), and $R^2$. **d** The Hill coefficients for the RPA-dT100-binding studies were calculated from the hill-plot transform of the binding data. **e** Schematic of the FRET-based experiment using fluorescent versions of RPA. **f** Based on the relationship between DBD-A and Tri-C (Fig. 3c–f) we estimated a distance of 45 Å between the fluorescent label sites. **g**, **h** Stopped flow fluorescence measurements of **g** RPA$^{WT}$-DBD-D-Cy5 and RPA$^{WT}$-DBD-A-Cy3 and **h** RPA$^{S178D}$-DBD-D-Cy5 and RPA$^{S178D}$-DBD-A-Cy3 FRET complex formation on different lengths of ssDNA (dT$_{20}$–dT$_{97}$). No ssDNA controls are shown in gray (denoted dT$_0$). FRET signal is only produced only when ssDNA ≥ 45nt is present. **i**, **j** Stopped flow experiments and analysis of **i** RPA$^{WT}$-DBD-D-Cy5 and RPA$^{WT}$-DBD-A-Cy3 and **j** RPA$^{S178D}$-DBD-D-Cy5 and RPA$^{S178D}$-DBD-A-Cy3 as a function of increasing length of ssDNA (nucleotides, n). Data were fitted with a two-step model and the observed rate of association (RPA–ssDNA or RPA–RPA association) plotted against nucleotide number. **i** The rate of association ($k_{obs,1}$) for RPA$^{WT}$ remains consistent with increasing ssDNA length. **j** An increase in rate of association ($k_{obs,1}$) correlated with an increase in ssDNA length is observed for the RPA$^{S178D}$ variant and suggests a cooperative-binding behavior of this mutant. Data points are an average of three experiments and are shown with SEM. Source data are provided as a Source Data file

the Tri-C that we assign as DBD-B. Multiple reconstructions from subclasses show that this domain possesses significant flexibility precluding its accurate reconstruction. However, the location of this domain is significantly different from the location of DBD-B (and thus DBD-A) in the crystal structure of the 30-nt compact mode[13], suggesting an alternative, extended, path for ssDNA. Models proposed by others[39], show that the DBD-A and/or DBD-B tandem binds to ssDNA in a more dynamic manner than previously thought, and our data suggest that these domains are stabilized when braced by another RPA on ssDNA as part of RPA–RPA interactions (see below). FRET-based analyses using labeled ssDNA to assess the configuration of the oligonucleotide when bound with RPA suggests that ssDNA is linearly arranged when bound by multiple ScRPA and that the extension is likely to be uniform within each ScRPA and between adjacent ScRPA molecules. What is more, even using a short dT$_{30}$ oligonucleotide capable of binding a single heterotrimer still shows a reduction in FRET, indicative of a linear conformation of ssDNA, as a result of stoichiometric binding of RPA, suggesting that the linear arrangement of ssDNA within RPA is the dominant configuration although we do not rule out the existence of the compact horse-shoe conformation observed in the crystal structure.

Our cryoEM model of two RPA complexes in close proximity on ssDNA indicates an RPA–RPA association and predicts that RPAs can contact one another via DBD-A of one RPA and DBD-E of the adjacent RPA. Our structures also predict that this interaction could stabilize a conformation that allows a linear path of ssDNA. We validated our hypothesis using a combination of FRET-based assays and protein domain interaction studies. Ensemble stopped-flow FRET experiments suggest a close association between two RPA molecules on ssDNA that is in line with predicted distances based on the association between DBD-E and DBD-A. Protein interaction experiments using purified

of these DBDs could differ in the context of the full-length protein as shown recently[39]. Our cryoEM data of ScRPA bound to dT$_{100}$, which we have shown is capable of accommodating at least three RPA molecules, show direct interactions between RPA molecules on ssDNA despite a range of flexible motions. We were able to reconstruct the relatively small Tri-C stably bound to a 20-nt segment of the ssDNA to surprisingly high resolution given its low molecular mass (~55 kDa). Both DBD-A and DBD-B were not easily discernable in these reconstructions, presumably due to their conformational flexibility relative to the Tri-C. In a small subset of the data we could resolve additional density underneath

DBD-A and DBD-E demonstrate a weak interaction between these domains. Furthermore, we used mutant proteins to validate the interface, which suggests a similar mode of interaction to that found in the fungal RPA crystal structure asymmetric unit.

A DBD-A–DBD-E interaction between two RPA molecules would certainly support cooperative ssDNA binding. Whilst we observe a physical interaction between RPA complexes, we do not observe in vitro cooperativity per se, which indicates that the interaction occurs after ssDNA association. This is further confirmed by the stopped flow experiments where ssDNA is required to observe a FRET signal and is also dependent on the length of the ssDNA. This is not surprising as the interactions between DBD-A and DBD-E are weak, and are only manifested when two RPA molecules are brought to close proximity by the ssDNA interactions. We hypothesized that a Mec1/Tel1-dependent phosphorylation site (Ser178) that sits close to the proposed RPA–RPA interaction interface may modulate the RPA–RPA interaction and thus cooperative behavior of RPA on ssDNA. In our DBD-A–DBD-E interaction studies the presence of a phosphomimetic mutation (S178D) produces an ~three-fold increase in affinity between these domains. Investigating the phosphomimetic mutant in the context of the full complex shows that ScRPA$^{S178D}$ can interact with ssDNA cooperatively although with a reduced affinity compared to wildtype. Analysis using fluorescent versions of RPA, suggests that DBD-A–DBD-E interactions are more stable, in line with our domain interaction data. However, the reduced coverage of ssDNA by DBD-A or DBD-A/B results in more flexibility between the two RPAs and would expose enough ssDNA to allow the action of recombinases and nucleases, supported by our data showing that the mutant RPA is more readily displaced by RecA from ssDNA and the ssDNA has slightly increased sensitivity to nucleases. We propose that the cooperative binding behavior of RPA could be 'switched on' by phosphorylation and this could be important for its efficient assembly and its subsequent recruitment and exchange with other ssDNA-binding factors such as Rad51.

In conclusion, our model shows how multiple RPAs could assemble on ssDNA and form higher order assemblies that modulate its function (summarized in Fig. 7). Our data also show that DBD-E (Rfa3/RPA14) is integral to this interaction, providing this subunit with an additional function, which contributes to explaining its lethality to the organism if deleted[45]. Interestingly, deletion of the *Schizosaccharomyces pombe* DBD-E ortholog, Ssb3, is dispensable for survival but results in the cells being sensitive to a number of DNA damaging agents, in particular those that disrupt DNA replication[46]. The interaction between DBD-A–DBD-E likely plays an important role and we propose that this interaction is post-transnationally modified to regulate RPA loading and/or exchange. However, this proposal and the exact mechanism of how this and other modifications affect its cellular activities require further in vitro and in vivo work in the context of RPA-containing cellular pathways, such as DNA damage response, homologous recombination, and DNA replication.

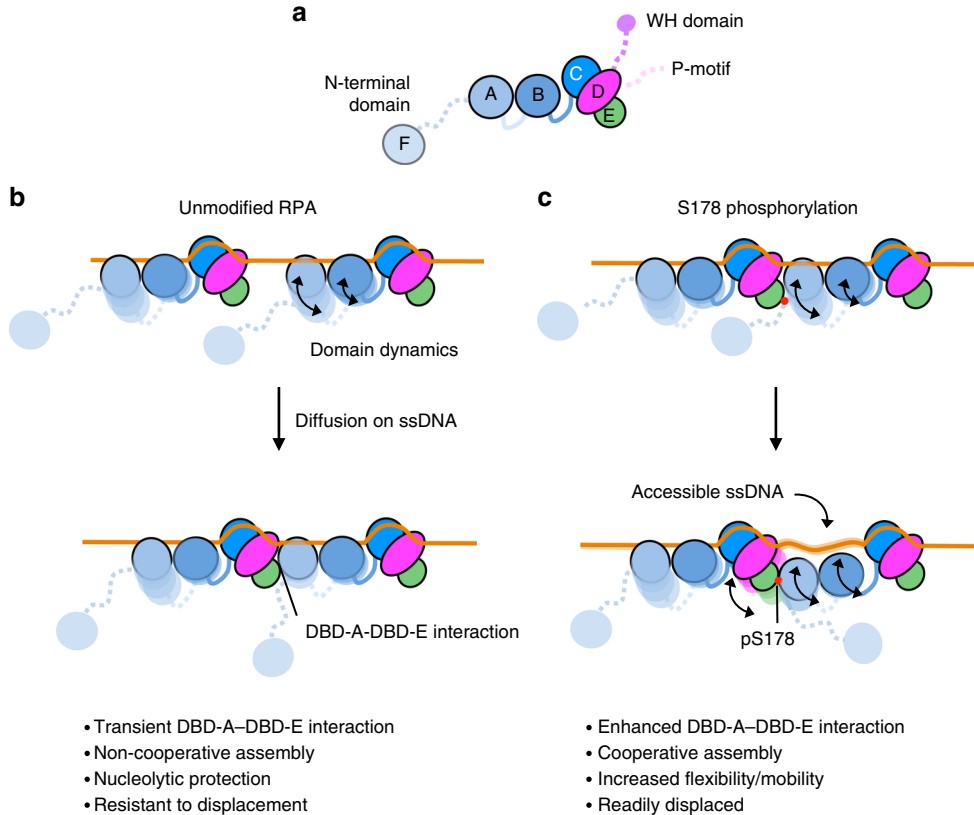

**Fig. 7** Proposed model of DBD dynamics when multiple RPAs are associated with ssDNA. **a** Schematic of yeast RPA (ScRPA) illustrating the DNA-binding domains (DBDs) F, A, B, C, D. Subunits of the RPA heterotrimer (Rfa1, Rfa2, and Rfa3) are colored as in Fig. 2. The winged-helix domain (WH) and phosphorylation motif (P-motif) of Rfa2 are also shown for completeness. **b** Unmodified RPA binds to ssDNA tracts with varying conformational states. DBD-A and DBD-B are dynamic, whereas the Tri-C is more stable. Through diffusion on ssDNA RPAs can interact with each other via an interaction between DBD-A and DBD-E. This interaction allows RPA to pass ssDNA from one molecule to another. **c** An S178D phosphomimetic mutant RPA has an enhanced DBD-A–DBD-E interaction that results in cooperative assembly of RPA on ssDNA. The positioning of DBD-A may not be associated with ssDNA and this allows easier access of the nucleic acid to other processing factors such as Rad51

## Methods

**DNA constructs.** The RPA heterotrimer expression vector was constructed using synthesized *RFA1*, *RFA2*, and *RFA3* genes, codon-optimized for expression in *Escherichia coli* (GeneArt, Invitrogen). Rfa1 (*RFA1*) was cloned in-frame into the MCS1 of a pRSF-Duet-1 to give a TEV-cleavable His6 tag, as part of the gene synthesis service (GeneJet). Both Rfa2 (*RFA2*) and Rfa3 (*RFA3*) were cloned into pMA after synthesis (GeneArt) and a DNA fragment containing both genes was excised by restriction endonucleases NdeI and AvrII (NEB), purified by gel extraction, and ligated using T4 ligase (NEB) into a pre-digested and purified pRSF-Duet-1-*Rfa1*. The resulting plasmid contained Rfa1 in MCS1 and Rfa2/Rfa3 in MCS2. The final expression construct was sequence verified before use.

To generate RPA domains for interactions studies, DNA fragments encoding residues 178–294 from Rfa1 and 1–99 from Rfa3 were amplified using the RPA expression plasmid (or the mutants) as a template and Phusion high fidelity DNA polymerase master mix (NEB) following the manufacturer's instructions. Primers used are listed in Supplementary Table 1. The DNA fragments were purified and cloned into a pOPINJ vector (a kind gift from Ray Owens, Addgene plasmid 26045) using In-Fusion enzyme (Clonetech, Takara) according to the manufacturer's instructions and ref. [47]. The lowercase sequence in the primers refers to the homologous region required for In-Fusion cloning into pOPINJ. The plasmids were sequence-verified prior to their use.

**Mutagenesis.** The DBD-A and DBD-E point mutants were generated using inverse PCR-based directed mutagenesis using CloneAmp DNA polymerase master mix (Takara) and the RPA complex expression vector as a template. The oligonulceotides used are listed in Supplementary Table 1. Mutagenesis was achieved by PCR using CloneAmp polymerase mix (Takara) and the manufacturers' instructions with an annealing temperature of 55 °C. Products were DpnI (NEB) treated to remove template and DNA purified using PCR purification (GeneJet, Thermo), followed by In-Fusion (Takara) before transformation into chemically competent NEB10-Beta *E. coli* (NEB). Clones were cultivated on LB agar supplemented with kanamycin (35 mg/l). The DNA from several clones were purified and checked by sequencing for the introduction of the mutations. The phosphomimetic Rfa1-S178D mutant was generated using PCR-based site-directed mutagenesis using Pfu Turbo DNA polymerase (Stratagene) and the RPA expression vector as a template. The oligonulceotides used are listed in Supplementary Table 1. Mutagenesis PCR was achieved using 18 cycles of 95 °C for 1 min; 95 °C for 50 s; 55 °C for 50 s; 68 °C, 9 min, with a final cycle with extension time of 7 min. The products were Dpn1 (NEB) treated to remove parental methylated template and checked by agarose gel electrophoresis before being transformed into cloning efficiency DH5α *E. coli*. Clones were cultivated on LB agar supplemented with kanamycin (35 mg/l). The DNA from several clones were purified and checked by sequencing for the introduction of the mutation.

Single-stranded poly-T oligonucleotides were purchased from IDT, as either 5′ 6-FAM or 5′ Cy5 $dT_{100}$ or unmodified $dT_{100}$ as well as 5′Cy5-$dT_7$.

**Expression and purification of RPA.** The RPA expression vector (or mutants of) was transformed into BL21 (DE3) *E. coli*. Transformants were selected using kanamycin and a single colony per liter was used to inoculate LB supplemented with kanamycin (34 mg/l) and incubate overnight at 37 °C without shaking. Once an $OD_{600\,nm}$ between 0.1 and 0.3 was reached the culture was shaken at 190 rpm for several hours until an $OD_{600\,nm}$ of 0.5–0.8. RPA expression was induced by the addition of 0.3 mM (final) IPTG at 37 °C for 3 h. Cells were harvested by centrifugation at 5000×*g* and either frozen at −80 °C until further use or were re-suspended in J0 buffer (30 mM HEPES pH 7.8, 0.5% *myo*-inositol, 0.02% Tween-20) supplemented with 500 mM NaCl and lysed by sonication. The lysate was clarified by high-speed centrifugation for 1 h at 20,000×*g*. The clarified lysate was filtered using a 0.45 μm syringe filter prior to loading onto a pre-equilibrated (with J0 with 500 mM NaCl) Ni-NTA column. Once the sample was loaded into the NiNTA, the column was washed with 10 column volumes (CV) of equilibration buffer (J0, supplemented with 500 mM NaCl), followed by 10CV of wash buffer (J0 supplemented with 750 mM NaCl and 40 mM Imidazole) and finally 10CV of J0. RPA was eluted with elution buffer (J0 supplemented with 250 mM Imidazole pH 8.0). The fractions containing RPA were pooled and dialyzed overnight at 4 °C in J0 buffer. The dialyzed RPA solution was further purified by anion exchange. The sample was loaded onto a HiTrap Q column (GE healthcare) equilibrated with J0 buffer and was washed sequentially with 1CV J0 buffer containing 50 mM KCl and 1CV of J0 supplemented 100 mM KCl. The protein is eluted with J0 buffer containing 400 mM KCl. At this point RPA was assessed as >95% pure by SDS–PAGE. Finally, RPA was polished by gel filtration (S200 16/60) in 20 mM HEPES, pH 7.8, 1.5 M NaCl, 2 mM DTT. The peak fractions containing RPA were pooled and concentrated by ultrafiltration together with buffer exchange, to remove the high concentration of NaCl, resulting in concentrated protein in 20 mM HEPES, pH 7.8, 300 mM NaCl, 1 mM DTT, and 0.5% glycerol. The presence of contaminating endogenous *E.coli* DNA was monitored by the absorbance ration between 260 nm/280 nm, which was ~0.6 for the purified RPA proteins. Protein was flash frozen in liquid nitrogen and stored at −80 °C until use.

**Expression and purification of fluorescent variants of RPA.** The plasmid expressing all three subunits of RPA (a kind gift from Dr. Marc S. Wold, University of Iowa) was modified to substitute the two original Amber stop codons with Ochre stop codon (TAA) using Q5 site-directed mutagenesis (EANP-scRPA-70-32-14). To generate plasmid for purification of RPA-Dbd-A⁴ᴬᶻᴾ, Threonine 211 at RPA 70 was substituted with TAG stop codon and a 6x poly-histidine tag was incorporated at the C-terminus of RPA 70 (EANP-scRPA-70ᴬ⁻⁴ᴬᶻᴾ-32-14). Similarly, to generate plasmid for purification of RPA-Dbd-D⁴ᴬᶻᴾ, Tryptophan at position 101 of RPA32 was substituted with TAG stop codon and a 6x poly-histidine tag was incorporated at the C-terminus of RPA 32 (EANP-scRPA70-32⁴ᴬᶻᴾ-14). Serine at position 178 of RPA 70 was substituted with aspartic acid (S178D) using Q5 site-directed mutagenesis. EANP-scRPA-70ᴬ⁻⁴ᴬᶻᴾ-32-14 and EANP-scRPA-70-32⁴ᴬᶻᴾ-14 were used as template plasmids to generate RPA-S178D-DBD-A⁴ᴬᶻᴾ and RPA-S178D-DBD-D⁴ᴬᶻᴾ, respectively.

Appropriate RPA plasmid (mentioned above) was cotransformed with pDule2-pCNF in BL21Ai cells. All RPA variants were expressed and purified to near homogeneity using immobilized metal affinity chromatography, anionic exchange using Q-sepharose and Heparin columns, sequentially, and finally by size-exclusion chromatography (SEC) using a Superdex S200 column[48]. All chromatographic steps were performed at 4 °C.

**Labeling proteins with fluorophores.** RPA variants carrying 4-AZP were labeled with Cy3 or Cy5[39,48]. Approximately 3 ml of RPA⁴⁻ᴬᶻᴾ (10 μM) was incubated on a rocker with a 1.5-fold molar excess (15 μM) of either DBCO-Cy3 or DBCO-Cy5 for 2 h at 4 °C. Labeled RPA variants were separated from excess dye using a Biogel-P4 gel filtration column (65 ml bed volume) using storage buffer (30 mM HEPES, pH 7.8, 100 mM KCl, and 10% (v/v) glycerol. Fractions containing labeled RPA were pooled, concentrated using a 30 kDa cut-off spin concentrator and flash frozen using liquid nitrogen. Fluorescent RPA were stored at −80 °C. Labeling efficiency was calculated using absorption values measured at 280 nm and $\varepsilon_{280} = 98{,}500\ \mathrm{M^{-1}\,cm^{-1}}$ for RPA, at 555 nm with $\varepsilon_{555} = 150{,}000\ \mathrm{M^{-1}\,cm^{-1}}$ for DBCO-Cy3, and at 650 nm with $\varepsilon_{650} = 250{,}000\ \mathrm{M^{-1}\,cm^{-1}}$ for DBCO-Cy5 fluorophores.

**Expression and purification of DBD-A and DBD-E.** Both GST-DBD-A and GST-DBD-E, and mutants thereof, were expressed in BL21 (DE3) *E. coli*. Owing to the constitutive overexpression of these constructs in *E. coli*, a single colony of transformants was picked and inoculated into a liter of LB supplemented with ampicillin (50 mg/l) and incubated at 37 °C without shaking overnight. The following morning the cultures were agitated at 180 rpm until mid-logarithmic phase ($OD_{600\,nm} = 0.5$–$0.8$) and the temperature reduced to 20–16 °C and incubated overnight with shaking. Cells were harvested by centrifugation at 5000×*g* and were frozen at −80 °C until further use. Cells were re-suspended in modified Tris-buffered saline (TBS; 50 mM Tris–HCl, 150 mM NaCl, 0.5 mM TCEP, pH 7.4) and lysed by sonication. Given that these constructs are 3C-cleavable GST fusions, proteins were purified using glutathione sepharose 4B (GE Healthcare) and standard procedures. Once the GST-fusion proteins were adsorbed, the domains were liberated using on-column 3C cleavage overnight at 4 °C and the resulting eluent was further purified by SEC at 4 °C. The protein was concentrated and stored in 20 mM Tris, pH 7.4, 150 mM NaCl and was either used immediately or flash frozen in liquid nitrogen for storage at −80 °C.

**Preparation of RPA–ssDNA nucleoprotein complexes.** Purified ScRPA was mixed with $dT_{100}$, resuspended in water, at a molar ratio of 5:1 for RPA:DNA and incubated on ice for 20 min. The RPA–ssDNA complexes were purified away from free RPA by SEC using a Superose 6 (10/30) in 20 mM Tris–HCl, 150 mM NaCl, 1 mM TCEP, pH 7.4 at 4 °C. We monitored two wavelengths, 260 and 280 nm, to assess the protein and DNA content of the elution peaks.

**SEC-MALS.** For SEC protein samples (two times the loop volume, 200 μl) were injected onto either a Superose 6 increase column (10/30 GE Healthcare) for RPA–ssDNA complexes or Superdex S200 (10.30 GE Heathcare) for Apo-RPA, mounted on a high-pressure liquid chromatography system (1260 Infinity; Agilent). Protein samples were taken from the central peak fraction of a previous preparative gel filtration and were resolved using 20 mM Tris–HCl, pH 7.5, 150 mM NaCl, 0.5 mM TCEP at 25 °C. Real-time light scattering and refractive index were simultaneously measured (Helios-II, T-rEX; Wyatt). The Astra software package was used for data analysis (Wyatt).

**Electron microscopy grid preparation.** Grids for cryo-EM data acquisition were prepared by depositing 3 μl of freshly purified RPA-$dT_{100}$ complex (concentration ~0.5 mg/ml), in 20 mM Tris–HCl, 150 mM NaCl, 1 mM DTT, pH 7.4) onto glow-discharged holey-carbon Quantifoil R2/2 copper grids. Samples were vitrified in liquid ethane at liquid nitrogen temperature using a Vitrobot Mk IV (FEI) set with a waiting time of 30 s and a blotting time of 1 s. Plunge freezing was performed at 4 °C and 100% humidity.

**CryoEM data collection**. For the RPA$^{WT}$-dT$_{100}$ complexes, a set of 1300 micrographs (dataset 1), fractionated into 24 frames, were acquired at NeCEN (Leiden, Netherlands) on a Titan Krios microscope. For the RPA$^{S178D}$-dT$_{100}$ complexes, another set of 2700 micrographs (dataset 2), fractionated into 41 frames, were acquired at eBIC (Oxfordshire, UK) on a Titan Krios. For both datasets, the microscopes were operated at an acceleration voltage of 300 kV with the specimen at cryogenic temperatures (approximately −180 °C) with images recorded at 1–4 μm underfocus on a K2 Summit direct electron detector at a nominal magnification of ~130,000, resulting in a pixel size of 1.06 Å, and a cumulative total electron dose of 50 e$^{-}$/Å$^2$.

**CryoEM image processing**. Initially, the quality of the micrographs and their power spectrum was assessed by visual inspection. Any images with abnormal background, heavy contamination, thick ice, or low contrast were discarded. Individual movie frames of good micrographs were aligned and dose-weighted using MotionCor2[49]. CTF parameters were estimated with Gctf[50] for the summed micrographs and quality of the images assessed by visual inspection of the summed image alongside the power spectrum. Particle picking was performed with Gautomatch using class averages obtained from negative stain images collected in house, or from later 2D classes from a subset of manually selected particle dataset. Subsequent image processing was performed in Relion 2.1 (ref. [51]).

**RPA reconstructions**. Although the WT and S178D are subtly different, the Tri-C region is consistent between these datasets and therefore we use both datasets to reconstruct a high-resolution structure of the Tri-C of RPA. Approximately 600,000 particles were picked using Gautomatch from dataset 1 (WT) and were extracted in boxes 160 × 160 pixels. Images were binned twice and subjected to several rounds of 2D classification, removing junk and noisy particles after each round. Selection of the best 2D classes resulted in a set of ~310,000 particles, which were further processed by 3D classification into four classes using an ab initio model generated in CryoSPARC[52] and filtered to 60 Å as a start model. The particles that gave the most detailed classes were combined prior to 3D autorefinement. The particles that were used for the final refinement were re-extracted 120 × 120 boxes. The final reconstruction was refined using 180,723 particles to resolution of 5.8 Å using a soft mask and post-processing in Relion. For dataset 2 (S178D), ~1.3M particles were picked using gautomatch with templates reprojected from the previous reconstruction. The particles were extracted in boxes 120 × 1620 pixels. Images were binned twice and subjected to a round of 2D classification showing good classes similar to dataset 1. After removing junk and noisy particles after this step ~1M particles remained. Due to the large number of particles, we split the particle.star file into batches containing ~110K particles in each and subjected these to rounds of 3D classification (four classes per batch) using the previous model (filtered to 15 Å) as a reference. 3D classes with good features were selected and the particle.star files of each good 3D class joined in Relion resulting in a final set of ~161K particles. We refined this dataset and reached a global resolution of 5.5 Å after post-processing, which showed helices and recognizable features for this protein. We subsequently joined the two datasets together and performed a global refinement with a filtered mask with a soft edge. This resulted in a final reconstruction, after post-processing in Relion, with a global resolution of 4.7 Å, which has more clear secondary structure features and the occasional side-chain expected at this resolution. The reconstruction and refinement scheme is outlined in Supplementary Fig. 3. Global estimates of resolution, as well as local resolution, were calculated in Relion using the gold-standard Fourier shell correlation criterion (FSC = 0.143)[53].

The RPA$^{WT}$-dimer reconstruction followed a similar processing strategy as the Tri-C and is summarized in Supplementary Fig. 3. Briefly, particles were picked using Gautomatch and extracted in boxes 240 × 240 pixels. Several rounds of 2D and 3D classifications produced a single set of particles that was used for 3D refinement. Analogously to the Tri-C reconstruction a CryoSPARC ab initio model, filtered to 60 Å was used as a start model. A single class (class 1) containing 32,583 particles was refined in Relion to a resolution of 7.5 Å. Structural features were improved using a Local Agreement Filtering Algorithm for Transmission EM Reconstructions (https://github.com/StructuralBiology-ICLMedicine—personal communication Kailash Ramlaul and Christopher H. S. Aylett). Global estimates of resolution were calculated in EMAN using gold standard Fourier Shell Correlation Criterion (FSC = 0.143) and local resolution estimates using ResMap.

**Model building**. The *S. cerevisiae* RPA Tri-C 4.7 Å map showed clear secondary structure elements and β-barrels of OB-folds, which allowed the unambiguous rigid-body fitting of either human and fungal crystal structures. To interpret our density with a yeast model, we created several homology models of Rfa1 (DBD-A, DBD-B, and DBD-C), Rfa2 (DBD-D) using Swiss-model or Phyre2[54]. An Rfa3 (DBD-E) homology model was obtained using the Rosetta server[55]. The subunits of the human crystal structure (pdb 1L1O) were replaced with the yeast homology models by superposition. The poly-dT oligonucleotide was taken from pdb 4GOP, extended with additional nucleotides in COOT and merged with the protein subunit models. The Tri-C with ssDNA model was flexibly fitted to the density using MDFF[33] using NAMD and implemented in VMD. For MDFF the map was converted to Situs format using IMAGIC[56] and the initial structure and secondary structure restraints for NAMD prepared using the VMD. A 50 ps MDFF simulation was performed before the resulting energy minimized structure was further refined using 150 cycles of Jelly-Body refinement in Refmac5[34] implemented in CCPEM[35] and five cycles of real-space refinement in Phenix[36] imposing secondary structure restraints.

**Microscale thermophoresis (MST)**. MST experiments were performed using a Monolith NT.115 instrument (NanoTemper Technologies, Germany) at room temperature. MST-binding buffer (20 mM Tris, pH 7.5, 150 mM NaCl, 0.5 mM TCEP) was used for all experiments. In all cases premium treated capillaries were used and the experiment conducted at 25 °C. MST data was either analyzed in NanoTemper Analysis 1.2.101 software or GraphPad Prism. In either case the data were fitted with the Hill equation, with the Hill coefficient constrained to 1 where 1:1 binding is expected. Each experiment was technically repeated at least three or more times and the mean half effective concentration (EC$_{50}$) values were calculated with standard error (SE).

**MST with ssDNA**. RPA–ssDNA interaction studies used synthetic DNA oligos purchased from IDT with either a 6-FAM or Cy5 5′ fluorophore. The fluorescently labeled DNA, in binding buffer, was diluted to a concentration of 100 nM and mixed with an equal volume of a serial dilution series of RPA or the phosphomimetic mutant RPA$^{S178D}$ and incubated at room temperature for 20 min before loading into MST capillaries. For ssDNA binding to individual DBD-A domains, 100 nM of Cy5-labeled dT$_7$, diluted in MST-binding buffer, was mixed with equal volumes of a dilution series of DBD-A or DBD-A$^{S178D}$, also in MST-binding buffer. Single MST experiments were performed using 20–30% LED power and 80% MST power with a wait time of 5 s, laser on time of 30 s and a back-diffusion time of 5 s.

**MST of DBD-A–DBD-E interactions**. Purified DBD-E, or its mutants, was labeled via its free cysteine residues using Monolith NT protein-labeling Kit Blue-Cys NT-495 (NanoTemper) and purified according to manufacturers instructions. 50 nM of labeled DBD-E, in MST-binding buffer, was mixed with equal volumes of a dilution series of DBD-A or DBD-A$^{S178D}$ also in MST-binding buffer supplemented with 0.01% Tween-20 and 0.1 mg/ml BSA, to prevent non-specific binding, and incubated at room temperature for 20 min before loading into MST capillaries. Single MST experiments were performed using 80% LED power and 80% MST power with a wait time of 5 s, laser on time of 15 s and a back-diffusion time of 5 s.

**DNA electrophoretic mobility shift assay (EMSA)**. Binding assays were carried out in 20 μl. 50 nM of 5′ Cy5-labeled dT$_{100}$ in binding buffer (20 mM Tris, pH 7.5, 150 mM NaCl, 10 mM MgCl$_2$, 0.01% Tween-20) and was added to a serial dilution of RPA (protein concentration range, 5.8 μM–11 nM). A no protein control was also prepared. Binding reactions were incubated at room temperature for 30 min before mixing with 1 μl of bromophenol blue loading buffer. 5 μl of samples were loaded into a non-denaturing Tris–borate 5% polyacrylamide gel, pre-run for 1 h. Gels were run for 90 min at 100 V. The DNA was visualized using in-gel fluorescence using a ChemiDoc gel imaging system (Bio-Rad).

**FRET-based assay for RPA DNA extension**. FRET based-assays were used to monitor the effect of scRPA and scRPA$^{S178D}$ binding on DNA conformations using methods previously described[57]. A Cary Eclipse Fluorimeter was used to excite Cy3 and observe Cy3 and Cy5 emission. The Cy3 excitation wavelength was set to 530 nm and emission was measured at 565 nm. Cy5 excitation was caused by the energy transfer from Cy3 and its emission was measured at 660 nm. Excitation and emission slit widths were all set to 10 nm. Experiments were carried out in Reaction buffer (50 mM Tris–HCl, pH 7.5, 5 mM MgCl, 100 mM NaCl, 1 mM DTT, and 0.1 mg/ml BSA) at 25 °C.

The effect of RPA binding on DNA conformation was tested by measuring the starting FRET level of free FRET-labeled ssDNA substrates and titrating increasing concentrations of RPA. The Cy3/Cy5-labeled ssDNA oligonucleotides were synthesized by IDT (Coralville, IA) and are shown in Supplementary Table 1. The 5′ terminal Cy5 dye is attached to the hydroxyl group of the ribose, and the internal Cy3 and Cy5 dyes are incorporated into the backbone between the bases. 10 nM of each FRET substrate was added to Reaction buffer and starting FRET was measured. Increasing concentrations of scRPA and scRPA$^{S178D}$ were titrated into the reaction and FRET was monitored. For each FRET substrate the FRET levels start higher, as free DNA forms a compact flexible structure in our reaction buffer, bringing the fluorophores closer together. As scRPA or scRPA$^{S178D}$ is added FRET decreases as the fluorophores move away from each other due to DNA extension. The data for each point are presented as average ± standard deviation for at least three independent experiments. FRET data for the ScRPA binding to dT$_{30}$ and dT$_{90}$(5–35) and dT$_{90}$(11–41) were fit to two-segment lines to determine the binding stoichiometry ($X_0$). Binding of the ScRPA$^{S178D}$ to dT$_{30}$ was fit to a quadratic-binding equation, whilst the interaction between ScRPA and dT$_{90}$(1–30) was analyzed using a Hill equation and the assumption that there are three binding sites on the 90-mer oligonucleotide. All analyses were carried out using GraphPad Prism 7.03 software. The results of each fit are presented as a value ± fitting error.

**FRET analysis of cooperative DNA binding**. Stopped flow experiments were carried out on a SX-2000 instrument (Applied Photophysics, UK) and performed by premixing 40 nM RPA–DBD-A$^{Cy3}$ and 40 nM RPA–DDB-D$^{Cy5}$ in one syringe and rapidly mixing it with 40 nM ssDNA oligonucleotides from another syringe. Oligonucleotides of varying lengths were used [(dT)$_n$; $n$ = 20, 35, 45, 60, 70, 79, 97, or 140 nt]. Cy3 was excited at 555 nm, and the change in Cy5 emission was monitored using a 645 nm cut-off filter (Newport, CA). DNA-binding data for dT$_{20}$ through dT$_{97}$ were fit using a two-step model (Kaleidagraph, Synergy Inc., USA), and the data for dT$_{140}$ were fit with a one-step model[39].

**Reporting summary**. Further information on experimental design is available in the Nature Research Reporting Summary linked to this article.

## Data availability

All experimental cryoEM density maps of yeast ScRPA-dT$_{100}$ described in this paper have been deposited to the Electron Microscopy DataBank (EMDB) under the accession code EMD-4410 together with a Tri-C model, which was deposited into the Protein Data Bank (PDB) under accession code 6I52. All other relevant data are available in this article and its Supplementary Information files. Data underlying Figures 4–6 and Supplementary Figures 1-6, 8-10, 12, 14-16 is provided as a Source Data file. A reporting summary for this Article is available as a Supplementary Information file.

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

## Acknowledgements

Initial screening was carried out at Imperial College London Centre for Structural Biology EM facility. We would like to thank J. Ortiz and L. Renault at the Netherlands Centre for Electron Nanoscopy (NeCEN) and, A. Siebert and Y. Chaban at eBIC (proposal EM14769) where high-resolution data were collected. eBIC is funded by the Wellcome Trust, MRC, and BBSRC. We thank members of the Zhang lab and others in the Section of Structural Biology for helpful insights and discussions. This work is funded by a Wellcome Trust Investigator Award to X.Z. (098412/Z/12/Z) and the National Institutes of Health [7R15GM110671] to E.A.

## Author contributions

L.A.Y. and X.Z. designed the project. CryoEM sample preparation, image processing and structure determination was performed by L.A.Y. and R.J.A. Protein construct cloning, mutagenesis, and purification was performed by L.A.Y and J.A.K., with contributions from R.L.P. MST and EMSA experiments and analysis were performed by L.A.Y. FRET-based experiments were performed and analyzed by N.P., C.C.C., E.A. and M.S. L.A.Y. and X.Z. analyzed the results and wrote the manuscript with contributions from M.S. and E.A. and edits by all other authors.

## Additional information

**Competing interests:** The authors declare no competing interests.

**Journal peer review information:** *Nature Communications* thanks the anonymous reviewers for their contributionto the peer review of this work. Peer reviewer reports are available.

