## [Peer Review File · Nature Communications]

Reviewers' comments:

Reviewer #1 (Remarks to the Author):

The work by Yates et al describes the 5.8-Angstrom cryo-EM structure of the RPA trimerization core from *Saccharomyces cerevisiae* bound to ssDNA and a 9-Angstrom structure of two adjacent RPAs. Together with supporting biochemistry and a mutational analysis, the work provides structural insights on how RPA decorates ssDNA and could coordinate the recruitment of other proteins. The topic and the data presented are interesting and relevant, but despite this interest, important issues should be addressed.

Major issues

- I have some concerns related with the resolution of the two structures presented, around 6 and 9-Angstrom. I see no obvious reason why the resolution should not be improved and thus, make more solid conclusions. This is especially relevant for the structure of two adjacent RPAs. This is an important structure for the conclusions in the paper, but the structure was solved at only 9-Angstrom resolution. Conclusions for how two adjacent RPAs associate are reached after fitting DbdB and estimating the path of the ssDNA on a low-resolution structure.

32,583 particles were used for the structure of two adjacent RPAs. The authors have collected 1,300 micrographs, a fairly small data set. The authors could have increased the number of micrographs 2-4 times, thus improving resolution and making more solid conclusions.

- I am concerned that there could be some overestimation in the resolution. The background in the 2D averages of the RPA "dimers" is noisy and I would expect better-defined structural features for maps at 5.8 and 9-Angstrom resolution.

Resolution curve for the RPA core (Suppl. Figure 2) has an unusual shape. It is an almost perfect 1.0 FSC for a long stretch, and then falls very smoothly, and there is strange shape again after 5.8 Angstrom.

For this and the curve for the "dimeric RPA", it is important to show all other FSC curves provided by Relion, mask, unmasked and specifically the FSC of the randomized phase data set, since gold-standard resolution estimation between two independently determined maps may be influenced by masking, filtering and alignment of noise.

- For both structures, the local resolution estimation maps are also unusual. For an average resolution of 5.8 Angstrom, the maps are colored with maximum and minimum resolution of 6 and 5-Angstrom. This range is too narrow, and thus it is not possible to see if there are areas at resolution poorer than 6. The range of resolutions in the local resolution maps should be broader.

- Model building. In methods, it is unclear how they choose the "best-fitting" model. The authors indicate in Methods that the "best-model" was assessed by docking and visual inspection in Chimera. They should provide some objective indication of why/how some models were selected and others discarded for model building, and the impact of this selection in the final model provided.

The authors mention that the 3 subunits of RPA could be distinguished from the density of their OB-folds and subunit specific helices. Since the components of RPA contain several OB-folds and the resolution of the map is only around 6-Angstrom, it is not obvious for the reader how the 3 proteins could be identified. A better explanation (likely including figures) of how the model was built would help clarifying that the subunits of RPA were unambiguously identified.

- The tracing of the path of ssDNA in the RPA trimeric core and the "RPA dimers" should be explained better. This is important for the conclusions of the paper, and especially in the case of the dimer. This is solved at low resolution and it is not sufficiently well explained how the path was

determined and how much ambiguity there is. Maybe some of the interpretations of the map are beyond the resolution of the structure provided.

- The whole section on the effects of the S178D is interesting but the difference with the WT are subtle and thus conclusions are mostly indications, rather than a solid proof. This limits some of the conclusions presented in the model, in Figure 7.
- The DbdA-Rfa3 interaction experiments show a modest increase in the affinity of the interaction when using the DbdA S178D mutant. Due to the large errors in of the data this effect seems very subtle. A possible way of confirming these results could be to repeat the MST experiments labeling the DbdA protein instead of the Rfa3.
- The increased cooperativity shown in the ssDNA binding experiments using full length RPA shows data containing large errors. These results would be more convincing if they could be repeated labeling the protein and using unlabeled DNA. Additionally, after the results shown in this work, it cannot be ruled out the possibility of the Ddc2-RPA binding being the one causing the higher cooperativity. It would be good to investigate the binding of Ddc2 to RPA and RPA S178D in the absence of ssDNA.
- In several of the EMSA experiments (Figure 5 and Suppl Figures) there are mentions to “diffuse” vs “discrete” bands. The differences are too small, and the interpretation seems quite subjective.
- Similarly, the EMSA experiment in Figure 5b, when compared to Suppl. Fig 1a, mentions that the complex assembles at lower concentration, but again the differences are small.
- In the same experiment, in Figure 5b, the authors indicate with asterisks a higher order RPA-DNA species, but the intensity of these bands is small, and unclear why they disappear at higher concentrations.
- The phosphomimetic DbdA S178D mutant used in several experiments suggests the interesting possibility of phosphorylation of S178 playing a role in increasing the cooperative binding of RPA molecules. Having the RPA S178D mutant available and taking into account the flexibility issues that have limited the resolution of the RPA dimer EM reconstruction it would be valuable trying to obtain a reconstruction the RPA S178D mutant. If indeed this mutant binds with higher affinity and cooperativity to other RPA molecules it may show less flexibility and allow for a higher resolution structure. Even some analysis of 2D averages of RPA S178D vs WT could provide some support for the model proposed in the manuscript.
- After seeing the affinities shown for the ssDNA-RPA-Ddc2 complexes it would be interesting trying to visualize this complex under the microscope and validate the proposed model.
- The model in Figure 7 is too speculative. The model should mostly summarize the solid conclusions from the manuscript. Instead, it covers most of the speculations from the manuscript for which there is no solid data presented, such as the path of the ssDNA and the effects of S178 phosphorylation. These speculations are good to discuss in the text, but way they are presented in Figure 7, they seem solid conclusions from the manuscript.

Minor issues

- The authors use sharpening B-factors of -440 and -200. Which is the B-factor automatic estimate provided by Relion?
- Grammatical errors: Line 241 (“show”, rather than “shows”); Figure 6a (SEC MALS, not SEC MALLS)
- Figure 4b. Color codes are not indicated in the legend, only in panel 4c, which is confusing for the reader.
- Figure 4c. They indicate as Rfa3 what in the text is named as Rfa3-OB, and this is confusing for

the reader.

- Methods. Model building. Last sentence (line 554-555). Unclear what the authors mean.

Reviewer #2 (Remarks to the Author):

Yates et al report the cryoEM structures of the *S. cerevisiae* replication protein A (RPA) heterotrimer bound to single-stranded (ss) DNA. Various treatments of the cryoEM data allowed creation of models of the single heterotrimer structure or two adjacent heterotrimers bound to ssDNA. The single heterotrimer closely resembles previous structures of related RPAs. The structure of two adjacent RPAs, however, identifies novel electron density at the RPA-RPA interface that the authors suggest could be from the DbdA/DdbB domains of the Rfa1 subunit and the Rfa3 subunit. A direct interaction between the DbdA domain of the Rfa1 subunit and the Rfa3 subunit is demonstrated in vitro. Additionally, the authors show that a phosphomimic mutation of a Ser in Rfa1 may alter ssDNA binding by the RPA variant and its recruitment of a DNA-damage response protein (Ddc2) to RPA/ssDNA. The work is interesting and will be of interest to many who study eukaryotic genome maintenance reactions. However, several technical issues and overinterpretation detailed below severely limit the impact of the paper.

Specific issues:

(1) In places, the manuscript is not written well and it could use additional critical evaluation to maximize the strength of the report.

(2) For the MST binding data shown in Figure 4, saturation is not reached with any sample, which makes it impossible to determine 50% binding saturation. This makes the method purely qualitative and relative strengths of binding quite poorly defined.

(3) The use of variants that alter the modeled DdbA/Rfa3 interface do eliminate complex assembly in Figure 4, which provides some confidence that the authors have appropriately assigned the interface. However, experiments that demonstrate proper folding of the variants must be provided to exclude the possibility that the failure to form complexes is not due to misfolding.

(4) The authors state that the Ser178Asp DdbA domain variant interacts with Rfa3 with a "modestly increased affinity" (page 6, line 223). The binding data actually overlap on Figure 4 and, as indicated above, fitting of these data is not appropriate. This statement should be removed.

(5) More information on how ssDNA binding data were fit needs to be included in the methods section. What equations were fit? How were parameters determined?

(6) For the ssDNA binding results in Figure 5a, it appears that only two points actually separated wt RPA and Ser178Asp RPA ssDNA binding. These are ~8 nM and ~80 nM (can't tell the actual RPA concentration due to the compressed X-axis hash marks). This very modest difference underscores the importance of defining how data were fit (point (5)), especially since these difference are leading to dramatic differences in Hill coefficients.

(7) Why is the Hill coefficient for wt RPA 1? Given that the authors have argued that wt RPA trimers interact, one would have assumed that this would be >1.

(8) The EMSA in Figure 6d appears superficially similar to that in Figure 5b. What evidence is there that Ddc2N is really in the supershifted bands? It would be far better to titrate Ddc2N into fixed [RPA-ssDNA] samples.

(9) Figure 6c suffers from the same issues as described for ssDNA binding in comment (6), except

that in this case there are no points that separate mean measurements beyond their error. How can different Hill coefficients ever be determined given this lack of separation?

We are pleased to note that both reviewers view this work interesting and relevant to many studying genome maintenance reactions and we thank the reviewers for their constructive criticism and recommendations. We have taken on board most of their suggestions and have carried out extensive additional work including: 1) an improved cryoEM reconstruction of the RPA trimerisation at 4.7 Å (from 5.8 Å) and, 2) an improved cryoEM reconstruction of two adjacent RPAs on ssDNA to 7.5 Å (from 9 Å), 3) additional biophysical experiments to quantify the affinities and cooperativity, 4) Fluorescent Resonant Energy Transfer (FRET) experiments to study the dynamics and conformations of the RPA on ssDNA. Together these complementary data, now presented in this revised manuscript, provide compelling evidence for the observed novel interactions between two RPA molecules on ssDNA and the ssDNA path when multiple RPA molecules are assembled. Furthermore, we show that a key phosphorylation event promotes cooperativity of RPA assembling on ssDNA while maintaining/enhancing the conformational flexibility along the ssDNA. This arrangement is important for exchange as the ssDNA is more accessible to other processing factors such as Rad51. The conformational flexibility, we speculate, provides a more efficient recruitment platform for downstream DNA processing factors *via* increased surveillance. Detailed response/explanations to the reviewer's comments are below:

Major issues

- I have some concerns related with the resolution of the two structures presented, around 6 and 9-Angstrom. I see no obvious reason why the resolution should not be improved and thus, make more solid conclusions. This is especially relevant for the structure of two adjacent RPAs. This is an important structure for the conclusions in the paper, but the structure was solved at only 9-Angstrom resolution. Conclusions for how two adjacent RPAs associate are reached after fitting DbdB and estimating the path of the ssDNA on a low-resolution structure. 32,583 particles were used for the structure of two adjacent RPAs. The authors have collected 1,300 micrographs, a fairly small data set. The authors could have increased the number of micrographs 2-4 times, thus improving resolution and making more solid conclusions.

We have now improved the resolution of both reconstructions (see above). The dimeric model has been improved to 7.5 Å. However, despite a very large dataset on the phosphorylation mutant (based on the reviewers recommendations), the dimeric reconstruction remains at modest resolution. 2D classification reveals highly dynamic arrangements of the two RPA molecules along the ssDNA. Nevertheless, our cryoEM reconstructions of the dimer and trimeric cores place DBD-A and DBD-B (of Rfa1/RPA70) at the interface with the adjacent RPA, which contributes DBD-E (Rfa3/RPA14) at the interface. We agree with the reviewer that at this modest resolution, one cannot accurately model and fit individual domains. Fortunately DBD-A of one RPA contacts DBD-E of another RPA in the crystal packing of the published crystal structure (PDB code 4GOP). We have thus placed the structural model of DBD-A – DBD-E derived from crystallographic symmetry into our dimeric reconstruction, which fits well into the density (Fig 3 and Supplementary Fig 7). Mutagenesis of residues at the interface indeed displays defects in DBD-A-DBD-E interactions (Fig. 5). Our data clearly show that there is significant conformational flexibility between DBD-A and DBD-B, and between DBD-B and DBD-C (trimerisation core) (Fig 2f and g). Interestingly DBD-A and DBD-B, thought to work together as the major DNA binding determinants for RPA, are shown here to have significant conformational flexibility between them and they interact with DNA in a highly dynamic fashion (Fig 3). These observations are supported by FRET measurements

in this manuscript (Fig. 6) and have been seen by others (Pokhrel et al., 2018, accepted NSMB, preprint <https://www.biorxiv.org/content/early/2018/10/04/435636>).

- I am concerned that there could be some overestimation in the resolution. The background in the 2D averages of the RPA “dimers” is noisy and I would expect better-defined structural features for maps at 5.8 and 9-Angstrom resolution. Resolution curve for the RPA core (Suppl. Figure 2) has an unusual shape. It is an almost perfect 1.0 FSC for a long stretch, and then falls very smoothly, and there is strange shape again after 5.8 Angstrom.

For this and the curve for the “dimeric RPA”, it is important to show all other FSC curves provided by Relion, mask, unmasked and specifically the FSC of the randomized phase data set, since gold-standard resolution estimation between two independently determined maps may be influenced by masking, filtering and alignment of noise.

We now display all the FSC curves provided by Relion as suggested. Our improved maps allow more detailed features to be derived. See revised manuscript in describing these structures.

- For both structures, the local resolution estimation maps are also unusual. For an average resolution of 5.8 Angstrom, the maps are colored with maximum and minimum resolution of 6 and 5-Angstrom. This range is too narrow, and thus it is not possible to see if there are areas at resolution poorer than 6. The range of resolutions in the local resolution maps should be broader.

We have now displayed a wider local resolution range (4.5 Å – 6.5 Å for 4.7 Å map and 5 Å – 9 Å for 7.5 Å).

- Model building. In methods, it is unclear how they choose the “best-fitting” model. The authors indicate in Methods that the “best-model” was assessed by docking and visual inspection in Chimera. They should provide some objective indication of why/how some models were selected and others discarded for model building, and the impact of this selection in the final model provided.

We have updated the methods to make it clearer to the reader how the model was built, in short how we define the correct subunits was predominantly driven by the fitting of the known human RPA Trimerisation core structure (pdb 1L1O), which provided a good fit, and swapping the subunits for homology models of the yeast protein corresponding to the same regions. This was subjected to MDFF together with Refmac and Phenix refinements.

The authors mention that the 3 subunits of RPA could be distinguished from the density of their OB-folds and subunit specific helices. Since the components of RPA contain several OB-folds and the resolution of the map is only around 6-Angstrom, it is not obvious for the reader how the 3 proteins could be identified. A better explanation (likely including figures) of how the model was built would help clarifying that the subunits of RPA were unambiguously identified.

The trimerisation core reconstruction at 4.7 Å displays features that can clearly be attributed to each of the three DBDs that contribute to the trimerisation core (DBD-C of Rfa1, DBD-D of Rfa2 and DBD-E of Rfa3). We have explained this in text and

illustrated in Fig. 2. However to avoid confusion, we have specified that the 3 subunits of RPA within the trimerisation core could be distinguished after the fitting. In addition, we have updated the methods to make it clearer to the reader how the model was built.

- The tracing of the path of ssDNA in the RPA trimeric core and the “RPA dimers” should be explained better. This is important for the conclusions of the paper, and especially in the case of the dimer. This is solved at low resolution and it is not sufficiently well explained how the path was determined and how much ambiguity there is. Maybe some of the interpretations of the map are beyond the resolution of the structure provided.

We have now explained this in details (page 7-8). We used crystal structures of RPA-ssDNA as a guide, especially in maintaining the relationship between individual binding domains (DBDs) and ssDNA. Once we placed and fitted individual DBDs, the ssDNA trace is then fixed in place. Our dimeric model suggests a more extended ssDNA path that is significantly different from those observed in the crystal structure, which shows a horse-shoe shape. We have utilised FRET experiments to monitor the ssDNA conformations, which support an extended instead of bent conformation (Fig. 4).

Additional evidence for a different path for ssDNA within a single RPA also come from our sub-nanometer reconstructions of the Tri-C with DBD-B, which shows DBD-B in a different location as compared to that determined in the RPA crystal structure. This would suggest that ssDNA could be bound by DBD-B and alter the path from a horseshoe configuration to a more linear arrangement. This is strongly supported by FRET analysis on a short oligo (dT30) that clearly shows an increase in FRET-distance (decrease in FRET) when a single RPA binds (Supplementary Fig 5).

- The whole section on the effects of the S178D is interesting but the difference with the WT are subtle and thus conclusions are mostly indications, rather than a solid proof. This limits some of the conclusions presented in the model, in Figure 7.

We have now conducted extensive analysis of the DNA binding properties of the S178D RPA protein. Interestingly, we show the differences in ssDNA affinity are 4-fold, the cooperativity, measured by the slope of a hill plot derived from newly acquired MST binding data, shows a change from 0.9 to 1.7. These are further supported by FRET data showing a similar phenomenon on long substrates (S178D, Hill >1.6, and therefore cooperative). Additionally we have corroborated these observations using ensemble FRET analysis of RPA-RPA complexes using fluorescent versions of RPA that infer cooperative assembly of RPA-coated ssDNA (equal to or longer than 45nt) – all these data shows the S178D binds ssDNA cooperatively whereas WT does not. We have described these additional data in the revised manuscript.

- The DbdA-Rfa3 interaction experiments show a modest increase in the affinity of the interaction when using the DbdA S178D mutant. Due to the large errors in of the data this effect seems very subtle. A possible way of confirming these results could be to repeat the MST experiments labeling the DbdA protein instead of the Rfa3.

We have since repeated these experiments and have shown that the DBD-A-DBD-E (previously noted as DbdA-Rfa3 in the last MS) interaction increases by 3-fold when a S178D mutation is introduced (Figures 5 and 6 in new manuscript). For these experiments, we now show the change in normalized fluorescence for each

individual experiment, which have been fitted with the Hill equation in GraphPad Prism to calculate the EC50 so that the data fitting parameters are shown.

- The increased cooperativity shown in the ssDNA binding experiments using full length RPA shows data containing large errors. These results would be more convincing if they could be repeated labeling the protein and using unlabeled DNA. Additionally, after the results shown in this work, it cannot be ruled out the possibility of the Ddc2-RPA binding being the one causing the higher cooperativity. It would be good to investigate the binding of Ddc2 to RPA and RPA S178D in the absence of ssDNA.

We have since repeated these experiments and have shown RPA-dT₁₀₀ and RPA^{S178D}-dT₁₀₀ binding curves separately for clarity (Figure 6b, c). The data were fitted with the Hill equation and apparent hill coefficients are shown alongside the half-maximal binding (EC50) and the calculated Bmax with the R-squared fit of the data, which all show that the data are well fit. The errors bars (SE) are also much reduced. As the determination of the Hill coefficient is not accurate by simply fitting the hill equation, we have also plotted a linear hill plot of the data to show the difference in the slope of the hill plot that provides a more accurate determination of the hill-coefficient using many more data points (Fig. 6d). In the original manuscript and in this one we did not perform ssDNA-binding experiments in the presence of Ddc2-N and therefore the tethering of two RPAs together did not influence the cooperative phenomenon we observe. Whilst it would be interesting to assess the effect of Ddc2-N-RPA complexes on the assembly of RPA-ssDNA it is beyond the scope of our study, since our manuscript focuses on the effects of the mutation on the assembly and the stability of the assembly.

- In several of the EMSA experiments (Figure 5 and Suppl Figures) there are mentions to “diffuse” vs “discrete” bands. The differences are too small, and the interpretation seems quite subjective.

We have removed these data in favor of better MST analysis alongside FRET-based data.

- Similarly, the EMSA experiment in Figure 5b, when compared to Suppl. Fig 1a, mentions that the complex assembles at lower concentration, but again the differences are small.

See above

- In the same experiment, in Figure 5b, the authors indicate with asterisks a higher order RPA-DNA species, but the intensity of these bands is small, and unclear why they disappear at higher concentrations.

We have removed these data in favor of better MST analysis alongside FRET-based data to assess the relative stoichiometry and binding constants

- The phosphomimetic DbdA S178D mutant used in several experiments suggests the interesting possibility of phosphorylation of S178 playing a role in increasing the cooperative binding of RPA molecules. Having the RPA S178D mutant available and taking into account the flexibility issues that have limited the resolution of the RPA

dimer EM reconstruction it would be valuable trying to obtain a reconstruction the RPA S178D mutant. If indeed this mutant binds with higher affinity and cooperativity to other RPA molecules it may show less flexibility and allow for a higher resolution structure. Even some analysis of 2D averages of RPA S178D vs WT could provide some support for the model proposed in the manuscript.

We have taken on board the advice and have collected a larger dataset using RPA S178D mutant bound to dT₁₀₀. Interestingly, despite the increased cooperativity and affinity between two RPA molecules, the flexibility displaced between different domains remains comparable and indeed slightly enhanced compared to wildtype proteins. Detailed analysis indicates that there is substantial flexibility between DBD-B and the trimerisation core within one RPA as well as between DBD-A and DBD-B of one RPA, these flexibilities contribute significantly to the limited resolution of RPA dimer. And the RPA dimers do not become more rigid by the increased cooperativity and affinity between two RPA molecules. In fact, the S178D mutant cryoEM data indicate there might be increased flexibility between these domains (Supplementary Fig 13). To this end, our biophysical and FRET data analysis indicate that whilst the DBD-A-DBD-E interaction is enhanced, the S178D mutant RPA possesses weaker ssDNA binding activity. We suggest that DBD-A may be disengaged from ssDNA whilst being bound by DBD-E and this increases the relative flexibility of the nucleoprotein complex. Additional experiments suggest that the S178D mutant protein assembled on ssDNA is more readily exchanged by RecA and the ssDNA is more sensitive to S1 nuclease (Supplementary Fig 16), suggesting that ssDNA is more exposed whereas RPA-RPA interactions are strengthened. The functional significance of this unexpected observation requires further investigations.

- After seeing the affinities shown for the ssDNA-RPA-Ddc2 complexes it would be interesting trying to visualize this complex under the microscope and validate the proposed model.

We agree but we feel this is beyond the scope of the current manuscript, which focuses on RPA assembly on ssDNA, which then acts as a recruitment platform for many downstream factors including Ddc2. We have therefore removed the data.

- The model in Figure 7 is too speculative. The model should mostly summarize the solid conclusions from the manuscript. Instead, it covers most of the speculations from the manuscript for which there is no solid data presented, such as the path of the ssDNA and the effects of S178 phosphorylation. These speculations are good to discuss in the text, but way they are presented in Figure 7, they seem solid conclusions from the manuscript.

We have modified the figure to represent a model based on cryoEM structural mode, biophysical and FRET analysis.

Minor issues

- The authors use sharpening B-factors of -440 and -200. Which is the B-factor automatic estimate provided by Relion?

B-factors used for sharpening the maps were initially determined using Relion, but have been manually selected based on the appearance of features consistent with the estimated resolution in the resulting maps.

- Grammatical errors: Line 241 (“show”, rather than “shows”); Figure 6a (SEC MALS, not SEC MALLS)

We have updated the manuscript significantly and where we have described the technique we have referred to it as “SEC-MALS”.

- Figure 4b. Color codes are not indicated in the legend, only in panel 4c, which is confusing for the reader.

We have updated the legends to include color codes

- Figure 4c. They indicate as Rfa3 what in the text is named as Rfa3-OB, and this is confusing for the reader.

For consistency with the RPA field and to avoid confusion regarding species-specific terms (RPA14, Rpa14 or Rfa3) we have now termed Rfa3 as DBD-E.

- Methods. Model building. Last sentence (line 554-555). Unclear what the authors mean.

We have updated the methods section so that it is clearer.

Reviewer #2 (Remarks to the Author):

Yates et al report the cryoEM structures of the *S. cerevisiae* replication protein A (RPA) heterotrimer bound to single-stranded (ss) DNA. Various treatments of the cryoEM data allowed creation of models of the single heterotrimer structure or two adjacent heterotrimers bound to ssDNA. The single heterotrimer closely resembles previous structures of related RPAs. The structure of two adjacent RPAs, however, identifies novel electron density at the RPA-RPA interface that the authors suggest could be from the DbdA/DdbB domains of the Rfa1 subunit and the Rfa3 subunit. A direct interaction between the DbdA domain of the Rfa1 subunit and the Rfa3 subunit is demonstrated in vitro. Additionally, the authors show that a phosphomimic mutation of a Ser in Rfa1 may alter ssDNA binding by the RPA variant and its recruitment of a DNA-damage response protein (Ddc2) to RPA/ssDNA. The work is interesting and will be of interest to many who study eukaryotic genome maintenance reactions.

However, several technical issues and overinterpretation detailed below severely limit the impact of the paper.

Specific issues:

(1) In places, the manuscript is not written well and it could use additional critical evaluation to maximize the strength of the report.

We hope the substantially revised manuscript has improved this significantly.

(2) For the MST binding data shown in Figure 4, saturation is not reached with any sample, which makes it impossible to determine 50% binding saturation. This makes the method purely qualitative and relative strengths of binding quite poorly defined.

We now show the change in normalized fluorescence for each individual experiment, which have been fitted with the Hill equation in GraphPad to calculate the EC50 so that the data fitting parameters can be shown. This is to make it more clear to the reader how the apparent EC50 values have been calculated.

(3) The use of variants that alter the modeled DdbA/Rfa3 interface do eliminate complex assembly in Figure 4, which provides some confidence that the authors have appropriately assigned the interface. However, experiments that demonstrate proper folding of the variants must be provided to exclude the possibility that the failure to form complexes is not due to misfolding.

The purified DBD-E behaves as a globular protein that can be purified by size-exclusion chromatography and produces a symmetrical peak at an elution volume expected for a globular protein of this size. We have performed circular dichroism analysis of the DBD-E to demonstrate that the isolated domain retains its secondary structure. The mutants also run through size-exclusion chromatography and can be purified to homogeneity analogous to DBD-E, providing further evidence that the domains and proteins are properly folded.

(4) The authors state that the Ser178Asp DdbA domain variant interacts with Rfa3 with a “modestly increased affinity” (page 6, line 223). The binding data actually overlap on Figure 4 and, as indicated above, fitting of these data is not appropriate. This statement should be removed.

We have repeated the experiments (Figure 4 and Supplementary Fig 12) and have shown that the S178D mutation results in a 3-4-fold increase in affinity that is not affected by the presence of ssDNA.

(5) More information on how ssDNA binding data were fit needs to be included in the methods section. What equations were fit? How were parameters determined?

We have updated the methods section to include what equations were used to fit the data. We have also provided the fitting parameters in the figures themselves for clarity.

(6) For the ssDNA binding results in Figure 5a, it appears that only two points actually separated wt RPA and Ser178Asp RPA ssDNA binding. These are ~8 nM and ~80 nM (can't tell the actual RPA concentration due to the compressed X-axis hash marks). This very modest difference underscores the importance of defining how data were fit (point (5)), especially since these difference are leading to dramatic differences in Hill coefficients.

We have repeated and updated the binding data. We now show the two binding curves separately, with each experiment reaching saturation, alongside the fitting parameters from the Hill equation. We have also provided a linear hill plot of the data to show the marked difference in the slope (corresponding to the Hill coefficient) between the WT and S178D. The data show an increase in hill coefficient from 1 to 1.7 (Figure 6b-d). These data are also corroborated by FRET measurements, which show the S178D mutant associating with ssDNA with a Hill coefficient of >1.6

(Supplementary Figure 14). The cooperativity is also inferred by ensemble FRET analysis using fluorescent RPA variants (Figure 6).

(7) Why is the Hill coefficient for wt RPA 1? Given that the authors have argued that wt RPA trimers interact, one would have assumed that this would be >1 .

We would argue that RPA-RPA interactions for the wild-type would only occur after they are assembled on ssDNA as the interactions between RPAs in solutions are weak. Indeed our FRET data (Figure 6) suggest that RPA-RPA complexes only occur in the presence of ssDNA indicating that this weak interaction is secondary to the higher affinity ssDNA binding. An enhance RPA-RPA interaction, together with a weaker ssDNA interaction, would result in cooperative ssDNA binding and this is the case for the S178D mutant.

(8) The EMSA in Figure 6d appears superficially similar to that in Figure 5b. What evidence is there that Ddc2N is really in the supershifted bands? It would be far better to titrate Ddc2N into fixed [RPA-ssDNA] samples.

These data have now been replaced by MST and smFRET of RPA interacting with ssDNA only.

(9) Figure 63 suffers from the same issues as described for ssDNA binding in comment (6), except that in this case there are no points that separate mean measurements beyond their error. How can different Hill coefficients ever be determined given this lack of separation?

These data have now been replaced.

Reviewer #1 (Remarks to the Author):

The new experiments, the editing of the text, and the improvements in resolution, have solved my issues with this work.

Minor comments

- In Figure 2g, a domain is identified and assigned as DBD-B. I think at this stage of the manuscript is not formally possible to completely discard that the domain is either DBD-F or DBD-A. But I felt this is probably DBD-B after seeing the structure of the RPA dimer.

- An information that I did not find in the paper, and that it would add some valuable information to the reader is if the conformation of DBD-B in the RPA monomer (fig 2g) is similar to that in the dimer (fig 3).

Reviewer #2 (Remarks to the Author):

My concerns have been addressed in the revision and the manuscript is acceptable.